# Transcriptional regulatory divergence underpinning species-specific learned vocalization in songbirds

**Hongdi Wang**[1], **Azusa Sawai**[1], **Noriyuki Toji**[2], **Rintaro Sugioka**[1], **Yukino Shibata**[1], **Yuika Suzuki**[1], **Yu Ji**[1], **Shin Hayase**[1], **Satoru Akama**[3], **Jun Sese**[3,4], **Kazuhiro Wada**[1,2,5]*

1 Graduate School of Life Science, Hokkaido University, Sapporo, Japan, 2 Faculty of Science, Hokkaido University, Sapporo, Japan, 3 National Institute of Advanced Industrial Science and Technology, Tokyo, Japan, 4 Humanome Lab Inc., Tokyo, Japan, 5 Department of Biological Sciences, Hokkaido University, Sapporo, Japan

* wada@sci.hokudai.ac.jp

**Data Availability Statement:** All RNA-seq data were deposited in the DDBJ Sequence Read Archive (submission numbers DRA005548, DRA002970, and DRA008696).

## Abstract

Learning of most motor skills is constrained in a species-specific manner. However, the proximate mechanisms underlying species-specific learned behaviors remain poorly understood. Songbirds acquire species-specific songs through learning, which is hypothesized to depend on species-specific patterns of gene expression in functionally specialized brain regions for vocal learning and production, called song nuclei. Here, we leveraged two closely related songbird species, zebra finch, owl finch, and their interspecific first-generation ($F_1$) hybrids, to relate transcriptional regulatory divergence between species with the production of species-specific songs. We quantified genome-wide gene expression in both species and compared this with allele-specific expression in $F_1$ hybrids to identify genes whose expression in song nuclei is regulated by species divergence in either *cis-* or *trans-* regulation. We found that divergence in transcriptional regulation altered the expression of approximately 10% of total transcribed genes and was linked to differential gene expression between the two species. Furthermore, *trans-*regulatory changes were more prevalent than *cis-*regulatory and were associated with synaptic formation and transmission in song nucleus RA, the avian analog of the mammalian laryngeal motor cortex. We identified brain-derived neurotrophic factor (BDNF) as an upstream mediator of *trans-*regulated genes in RA, with a significant correlation between individual variation in BDNF expression level and species-specific song phenotypes in $F_1$ hybrids. This was supported by the fact that the pharmacological overactivation of BDNF receptors altered the expression of its *trans-*regulated genes in the RA, thus disrupting the learned song structures of adult zebra finch songs at the acoustic and sequence levels. These results demonstrate functional neurogenetic associations between divergence in region-specific transcriptional regulation and species-specific learned behaviors.

**Funding:** This work was supported by a Japanese MEXT scholarship and the China Scholarship Council (CSC#201408210091) to HW, MEXT/JSPS KAKENHI Grant Number #4903-JP17H06380, JP17H05932, JP17K19629, and JP18H02520 to KW, and RNA-seq experiments were supported by MEXT KAKENHI 221S0002. The funders had no role in study design, data collection and analysis, decision to publish, or preparation of the manuscript.

**Competing interests:** The authors have declared that no competing interests exist.

**Abbreviations:** AFP, anterior forebrain pathway; AM, amplitude modulation; Area X, Area X of the striatum; ASE, allele-specific expression; BDNF, brain-derived neurotrophic factor; CRMP, collapsin response mediator protein; DLM, dorsal lateral nucleus of the medial thalamus; FM, frequency modulation; FoxP2, Forkhead box protein P2; $F_1$, first-generation; GAD, glutamate decarboxylase; GO, Gene Ontology; GRIK1, Glutamate receptor, ionotropic, kainate type 1; GRIN, NMDA glutamate receptor; GTF, Gene Transfer Format; HTR1B, 5-hydroxytryptamine receptor 1B; indel, insert and deletion; IPA, Ingenuity Pathway Analysis; LMAN, lateral magnocellular nucleus of the anterior nidopallium; LMO7, LIM domain only protein 7; NGF, nerve growth factor; NPY, neuropeptide Y; nXIIts, tracheosyringeal part of the hypoglossal nucleus; OF, owl finch; OZ, first-generation hybrid offspring between owl finch female and zebra finch male; PCA, principal component analysis; RA, robust nucleus of the arcopallium; RAB5A, Ras-related protein Rab5A; RASGEF1B, Ras-GEF domain-containing family 1B; RIN, RNA integrity number; RNA-seq, RNA sequencing; RPKM, reads per kilobase of transcript per million reads mapped; SDE, species-differentially expressed; ss-SNP, species-specific SNP; SSM, syllable similarity matrix; TrkB, tropomyosin receptor kinase B; TTX, tetrodotoxin; ZF, zebra finch; ZO, first-generation hybrid offspring between zebra finch female and owl finch male; 7,8-DHF, 7,8-dihydroxyflavone.

## Introduction

Species-specific behavior plays a role in a variety of inter- and intraspecific interactions, including reproduction and habitat use, in which species differences are thought to be an important factor in species co-occurrence [1–3]. Such species-specific behaviors can arise via species differences in the structure and development of the neural circuits underlying behavior [4–6]. Differences between closely related species are thought to be driven by differential expression and functional changes of orthologous genes in conserved neural circuits, which are often in turn driven by transcriptional regulatory divergence [7–10]. Transcriptional regulatory divergence between species can arise due to species divergence in *cis*-regulatory elements that affect the transcriptional rate and stability, and/or in *trans*-regulatory factors that access *cis*-regulatory elements [11–16] (**Fig 1A**). However, it remains largely unknown how transcriptional regulatory divergence contributes to the generation of species-specific behavior, especially in the case of learned behavior.

Songs produced by oscine birds are complex vocal signals acquired through vocal learning [17,18]. Songs are species-specific, and these species differences play an important role in mating interactions and territory defenses within and between species [1,19,20]. In the songbird brain, a conserved neural circuit specialized for vocal learning, called the song system, contributes to song learning and production [18,21,22]. Birdsong is composed of two main traits associated with species specificity: the acoustic elements (syllables) and the temporal pattern (sequence) of song. The production of syllable acoustics and sequence is mainly regulated by the robust nucleus of the arcopallium (RA) and the song nuclei HVC (proper name), respectively, in the vocal motor circuit of the song system (**Fig 1B**) [22–24]. The importance of these song nuclei in determining species-specific song traits suggests an underlying causative role of species differences in the structure and activity of these regions. Consistent with this, a variety of genes, including transcription factors and neuromodulator receptors, are differentially expressed in these song nuclei between species, even in a laboratory-controlled environment [25–27]. However, a key gap in our knowledge is how species-specific patterns of gene expression in these regions arise via regulatory differences between species.

In this study, we used two closely related songbird species, zebra finch (ZF; *Taeniopygia guttata*), owl finch (OF; *T. bichenovii*), and their interspecific first-generation ($F_1$) hybrids, to elucidate how transcriptional regulatory divergence is associated with species-specific song (**Fig 1C**). These two species diverged about 6.5 million years ago and share overlapping habitats in the north and west of Australia [28,29]. In addition, they produce songs with characteristic species-specific syllable acoustics and sequence. By comparing the gene expression ratio between the two species and the allele-specific expression (ASE) ratio in the $F_1$ hybrids (**Fig 1D**), we assessed the total number of genes whose expression differs by divergence in *cis*- versus *trans*-transcriptional regulation between the two species. On the basis of Gene Ontology (GO) enrichment and the upstream regulatory analyses of transcriptional regulation–altered genes, we identified the candidate key upstream modulators of these differentially regulated genes and examined the functional effects of altered transcriptional regulation in the song nuclei.

## Results

### Species difference in song phenotypes between ZF and OF

First, we compared the song features of ZF and OF reared with conspecific song tutoring in our breeding colony to confirm whether a laboratory-controlled environment could maintain species-specific song features. We compared the songs of the two species regarding syllable

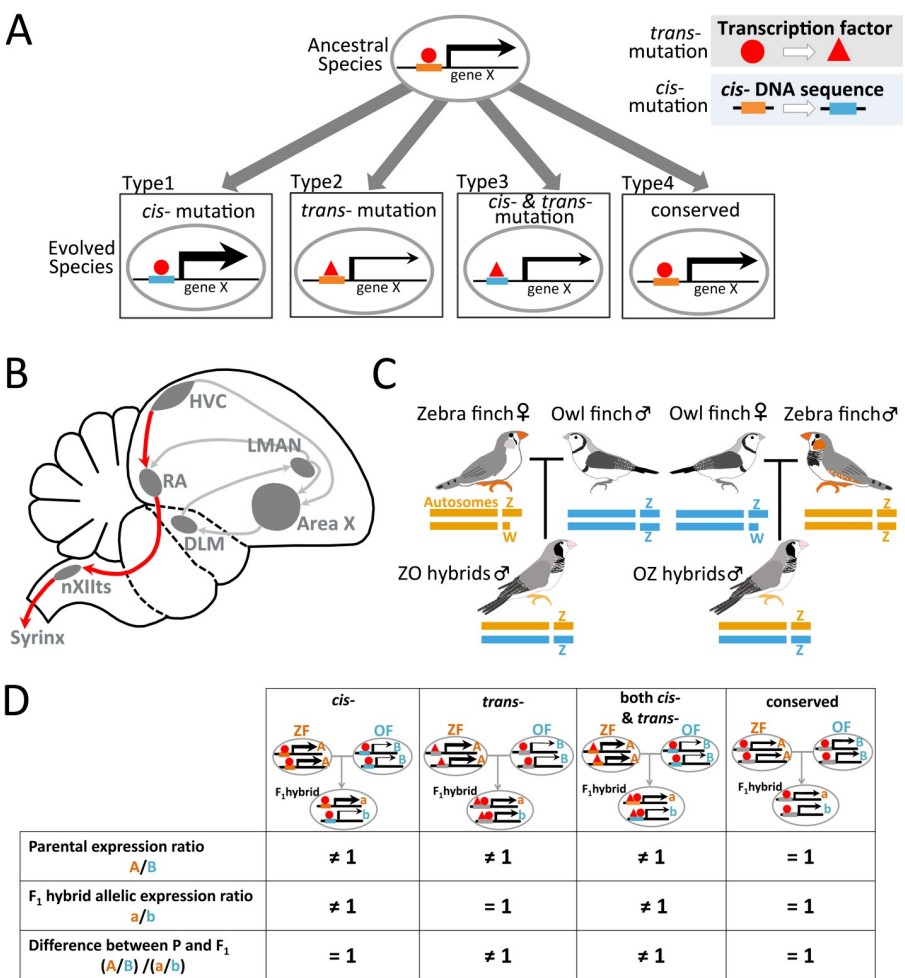

**Fig 1. *Cis-* and/or *trans*-regulatory changes during species differentiation. (A)** During evolution, *cis-* and/or *trans*-regulatory elements change gene expression levels between closely related species. (B) Schematic showing selected song-control regions and connections in the songbird brain. The posterior motor pathway and the anterior cortico-basal ganglia-thalamic circuit (anterior forebrain pathway [AFP]) are represented as red and gray lines, respectively. (C) Genome composition of reciprocal $F_1$ hybrids between zebra finch (ZF) and owl finch (OF). ZO represents $F_1$ hybrid offspring between ZF♀ and OF♂. OZ hybrids are the opposite. Male $F_1$ hybrids share identical sets of auto- and sex chromosomes. (D) Classification of species differences in *cis-* and/or *trans*-regulations based on the comparison of the relative gene expression ratio between parental species and the allelic expression ratio in their $F_1$ hybrids. For each gene, "A" and "B" represent gene expression levels in ZF and OF, respectively. "a" and "b" represent gene expression levels from ZF and OF alleles, respectively, in $F_1$ hybrids. "A/B" and "a/b" are the expression ratio between parental species and the allelic expression ratio in $F_1$ hybrids, respectively. Area X, Area X of the striatum; DLM, dorsal lateral nucleus of the medial thalamus; $F_1$, first-generation; HVC, used as a proper name; LMAN, lateral magnocellular nucleus of the anterior nidopallium; nXIIts, tracheosyringeal part of the hypoglossal nucleus; RA, the robust nucleus of the arcopallium.

acoustics and sequential features (12 parameters) at the adult stage (**Fig 2A**) and identified significant differences in six acoustic syllable parameters (i.e., syllable duration, inter-syllable gap duration, entropy variance, amplitude modulation [AM] variance, mean frequency modulation [FM], and FM variance) and in syllable sequence features (motif and repetition transition rates) ($n = 6$ birds each, $p < 0.01$, one-way ANOVA) (**Fig 2B and 2C** and **S1 Fig**) [30, 31]. We found that the range but not the pattern of each acoustic feature's distribution overlapped between ZFs and OFs (3,000 syllables from $n = 6$ birds each and 500 syllables/bird) (**S1 Fig**),

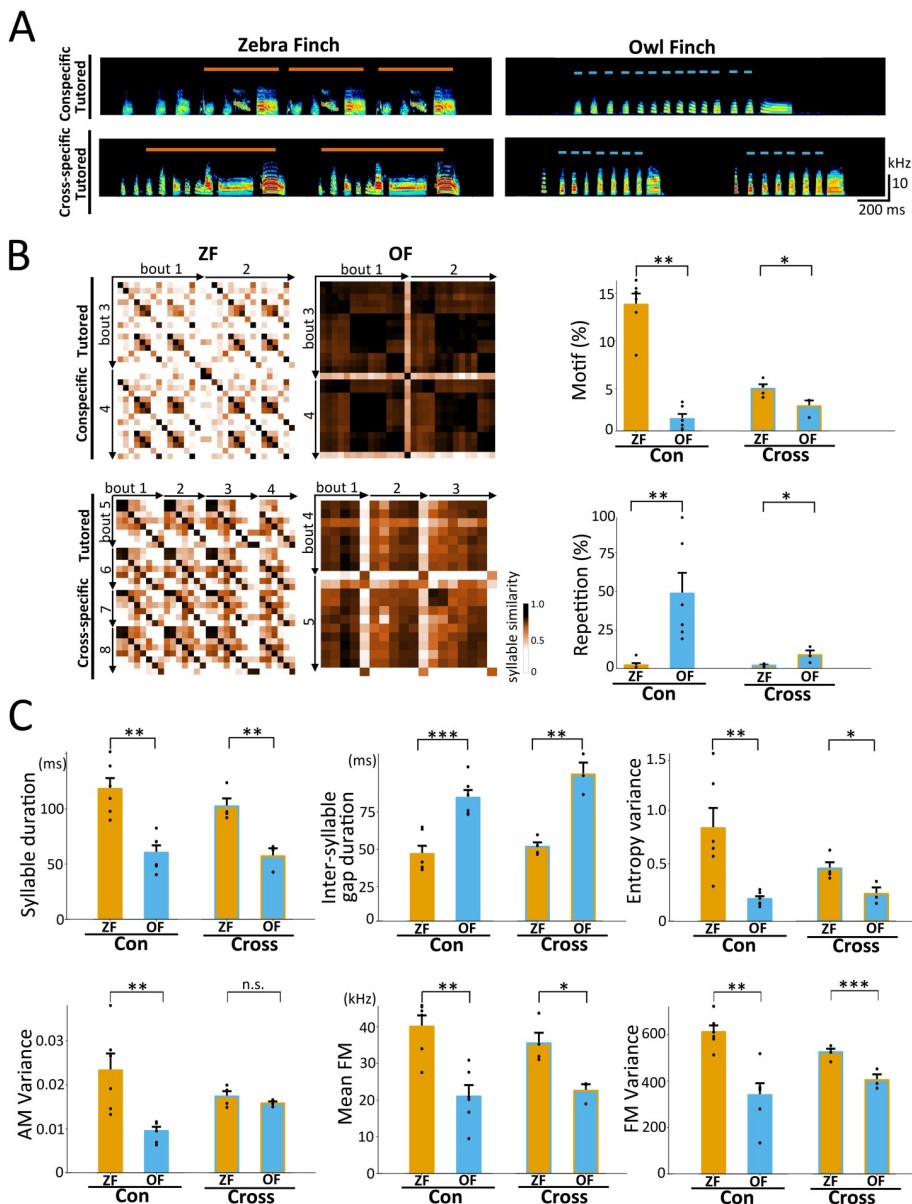

**Fig 2. Species difference in song structures between ZF and OF.** (A) Typical examples of songs from ZFs and OFs that were reared with conspecific song tutoring and cross-species song tutoring. Orange solid and blue dotted lines represent the motif and repetitive structure of syllables, respectively. (B) Species differences in the syllable sequence of ZF and OF songs. (Left) Syllable similarity matrices for songs produced by ZFs and OFs that were reared with conspecific song tutoring and cross-species song tutoring. (Right) Motif and repetition indices of ZF and OF songs (*n* = 6 each from conspecific song tutored ZF and OF, *n* = 4 and 3 from cross-species song tutored ZF and OF, respectively; one-way ANOVA, *$p < 0.05$, **$p < 0.01$). Each dot corresponds to an individual bird. (C) Species differences in syllable acoustics (syllable duration, inter-syllable gap duration, entropy variance, AM variance, mean FM, and FM variance) of ZF and OF songs ("Con": *n* = 6 each from conspecific song tutored ZF and OF; "Cross": *n* = 4 and 3 from cross-species song–tutored ZF and OF, respectively; one-way ANOVA, *$p < 0.05$, **$p < 0.01$, ***$p < 0.001$). Each dot corresponds to an individual bird. Relevant data values are included in **S1 Data**. AM, amplitude modulation; FM, frequency modulation; OF, owl finch; ZF, zebra finch.

thus suggesting that the species differences in the syllable acoustics were not caused by physical species-specific constraints in the peripheral vocal organs.

We further performed cross-species song tutoring experiments to examine how genetic and environmental factors contribute to generate species-specific song features of ZF and OF (**Fig 2A**). Under the cross-species song tutoring condition, juveniles heard only the counter-species songs as tutor songs. By comparing the *p*-values of song feature differences between conspecific and cross-species song tutoring conditions, we found that song tutoring affected most of the song parameters, including syllable sequence and acoustics (i.e., inter-syllable gap duration, entropy variance, AM variance, mean FM, and FM variance) (**Fig 2B and 2C**). However, except for AM variance, all song parameters retained species specificity (ZF, $n = 4$, OF, $n = 3$; one-way ANOVA, $p < 0.05$). In line with this result, we performed principal component analysis (PCA) to investigate the song feature distribution of conspecific and cross-species song tutored birds by reducing the dimensionality of the syllable acoustics and sequential features. We observed that clusters were separable by species but not by song tutoring conditions (**S1 Fig**). As many studies in songbirds reported [32–35], these results also indicate that song learning of these two species is implemented based on species-specific genetic constraint.

## Genome-wide transcriptional analysis between ZF, OF, and F$_1$ hybrids

We then conducted a genome-wide transcriptional analysis to elucidate divergence of transcriptional regulation between ZF and OF in their song nuclei. For this purpose, using laser microdissected HVC and RA tissues from ZFs and OFs, we identified 11,501 and 11,487 genes in HVC and RA, respectively, as genes with detectable expression levels in either ZF or OF (reads per kilobase of transcript per million reads mapped [RPKM] $\geq 1$). We then calculated the expression ratio between ZF and OF for each gene as "$A/B$" = RPKM$_{\text{(ZF average)}}$/RPKM$_{\text{(OF average)}}$ ($n = 4$ birds each) (**Figs 1D and 3A, S2 and S3 Figs**).

Based on a comparison of whole brain transcriptome between ZF and OF, a total of 2,409,063 SNPs were identified as species-specific SNPs (ss-SNPs) in their transcribed sequences. Using the ss-SNPs for the quantification of ASE ratios in the F$_1$ hybrids, we set a cutoff to extract genes with $\geq 5$ reads at each ss-SNP position and median RPKM $\geq 10$ ($n = 4$ each from ZO and OZ hybrids). Totals of 5,827 and 6,328 genes passed the criteria in HVC and RA, respectively. The ASE ratio of each gene in individual F$_1$ hybrids was calculated as "$a/b$" = Reads$_{\text{(ZF allele)}}$/Reads$_{\text{(OF allele)}}$ (**Figs 1D and 3B and S2 Fig**). To date, there is no evidence for paternal and maternal genomic imprinting in avian species [36]. In line with this, we identified no genes with a significant paternal or maternal bias in allelic expression in ZO and OZ hybrids. Furthermore, the two reciprocal F$_1$ hybrids (ZO and OZ) have an extremely high correlation in their ASE ratios (Pearson correlation coefficient, $r = 0.527$, $p < 2.2 \times 10^{-16}$ in HVC; $r = 0.550$, $p < 2.2 \times 10^{-16}$ in RA) (**S4 Fig**). Therefore, we treated ZO and OZ hybrids equally when calculating ASE ratios.

## Transcriptional regulatory divergence between ZF and OF

Transcriptional differences, *cis*- and/or *trans*-regulation, for each gene can be evaluated using the gene expression ratio between two species and the ASE ratio in the F$_1$ hybrids [14,16,37–39]. ASE in the F$_1$ hybrids reflects *cis*-dependent differences between the alleles of each parental species, because the two alleles of each gene are exposed to same *trans*-acting regulatory environment in cells. By comparing the gene expression ratio between parental species and the ASE ratio in F$_1$ hybrids, we determined the following five categories of transcriptional regulatory divergences: (i) "*cis*-regulation" for genes with significant *cis*- but not *trans*-effects (with $a/b \neq 1$ and $A/B = a/b$) as "*cis*-regulated genes," (ii) "*trans*-regulation" for genes with significant *trans*- but not *cis*-effects (with $a/b = 1$ and $A/B \neq a/b$) as "*trans*-regulated genes," (iii) "both *cis*- and *trans*-regulation" for genes with both significant *cis*- and *trans*-effects (with $a/b$

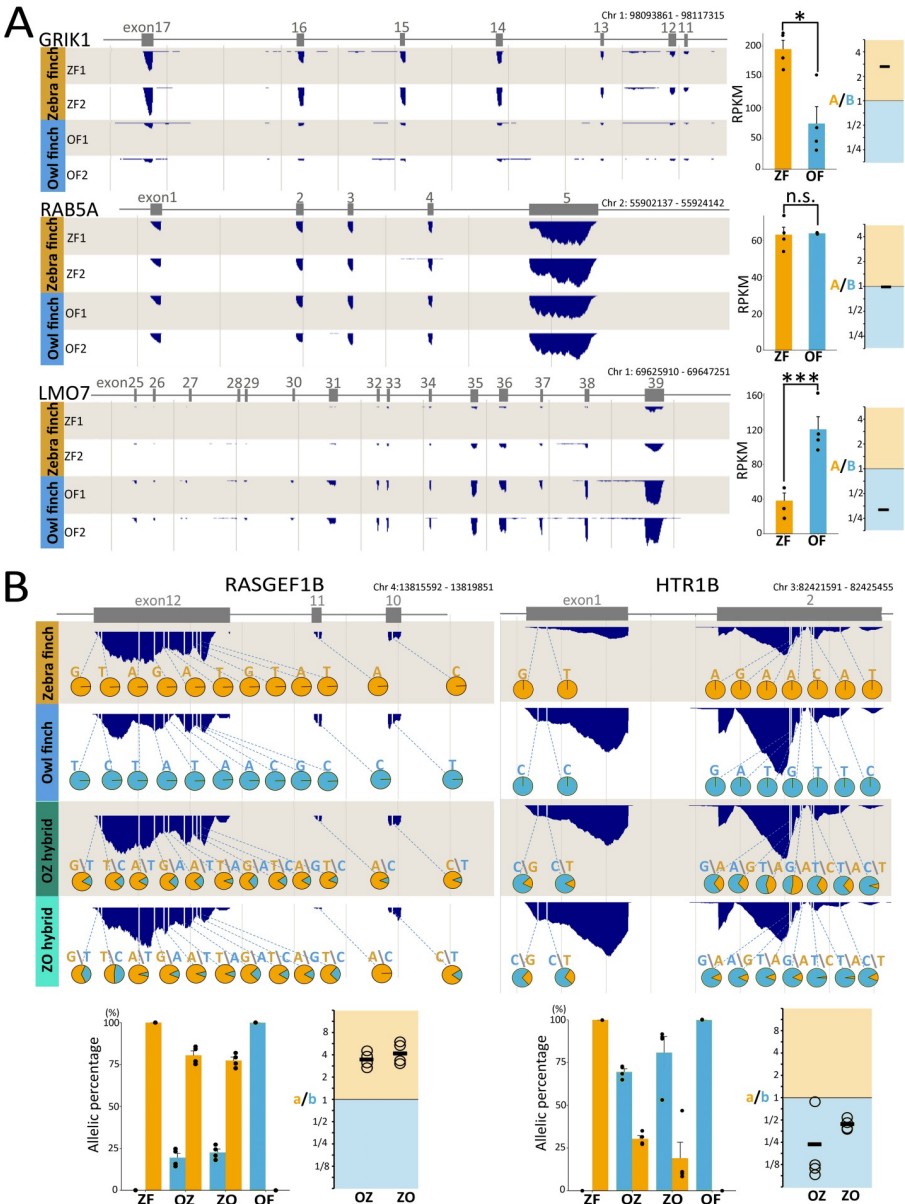

**Fig 3. Species differences in gene expression between ZF and OF and ASE in F₁ hybrids. (A)** Examples of species differences in gene expression between ZF and OF. (Left panels) Expression levels of *GRIK1*, *RAB5A*, and *LMO7* in song nucleus RA of ZFs and OFs. Gray boxes represent the position of exons for each gene. Dark blue peaks below exons represent read density. (Right panels) Gene expression levels in ZF and OF and the average of the expression ratio between ZF and OF. Each dot represents the RPKM value for individual. Mean ± SEM ($n$ = 4 birds each; one-way ANOVA, $^{*}p < 0.05$, $^{***}p < 0.001$, n.s., not significant). *RAB5A* is an example with no expression difference between ZF and OF. **(B)** Examples of ASE in F₁ hybrids. (Upper panels) Allelic expression ratios in F₁ hybrids at species-specific SNPs (ss-SNPs) of *RASGEF1B* and *HTR1B* in song nucleus RA. Dark blue peaks below exons represent read density. White bars in the dark blue–colored peaks represent ss-SNP positions. Pie charts of each ss-SNP represent the percentage of transcribed read numbers from ZF (orange) and OF (blue) alleles. (Bottom panels) The percentage and ratio of parental species-allelic expression of *RASGEF1B* and *HTR1B* in OZ and ZO F₁ hybrids. Each dot represents average allelic expression ratios of all ss-SNPs in one individual ($n$ = 4 birds each, mean). Orange- and blue-colored bars represent the values from ZF and OF alleles, respectively. Mean ± SEM ($n$ = 4 birds each). Relevant data values are included in **S2 Data**. ASE, allele-specific expression; Chr, chromosome; F₁, first-generation; *GRIK1*, Glutamate receptor, ionotropic, kainate type 1; *HTR1B*, 5-hydroxytryptamine receptor 1B; *LMO7*, LIM domain only protein 7; OF, owl finch; OZ, F₁ hybrid offspring between OF♀ and ZF♂; RA, robust nucleus of the arcopallium; *RAB5A*, Ras-related protein Rab5A; *RASGEF1B*, Ras-GEF domain-containing family 1B; RPKM, reads per kilobase of transcript per million reads mapped; ZF, zebra finch; ZO, F₁ hybrid offspring between ZF♀ and OF♂.

$\neq 1$ and $A/B \neq a/b$) as "both *cis*- and *trans*-regulated genes," (iv) "conserved regulation" for genes with no significant *cis*- or *trans*-effects (with $a/b = 1$ and $A/B = a/b$) as "conserved genes," and (v) ambiguous regulation (**Figs 1D and 4A and S2 Fig**). For this categorization of transcriptional regulatory divergence, we applied a cross-replicate comparison of ASE ratios in the $F_1$ hybrids, through which we could minimize incorrect estimation of *cis*- and artificial negative correlation in *cis*- versus *trans*-comparison (see Materials and methods) [40]. After this procedure, we observed that over 75% and 10% of the examined genes were expressed in both HVC and RA with either "conserved" or "ambiguous" regulation between ZF and OF, respectively (**Fig 4B**). In contrast, transcriptional regulatory divergence changed the expression of 158 (2.7% of the total 5,827 genes), 271 (4.7%), and 183 (3.1%) genes in HVC categorized as *cis*-, *trans*-, and both *cis*- and *trans*-regulated genes, respectively. Likewise, in RA, the expression of 246 (3.9% of the total 6,328 genes), 383 (6.1%), and 183 (2.9%) genes was altered by *cis*-, *trans*-, and both *cis*- and *trans*-regulatory changes between the two species, respectively (**Fig 4A and 4B**).

In both HVC and RA, *trans*-alteration was more prevalent than *cis*-alteration. These results indicated that the expression of 600–800 genes (approximately 10%–15% of the total expressed genes) in the vocal motor song nuclei was modified by altered transcriptional regulation between the two species. Furthermore, a majority of the genes under conserved regulation were highly expressed in both HVC and RA (3,523 genes of 4,489 [78.5%] and 4,782 [73.7%] genes expressed in HVC and RA, respectively). In contrast, most of the *cis*- and/or *trans*-regulated genes were not shared between HVC and RA (**Fig 4B**), showing a brain region–specific transcriptional regulatory alteration. Although this result was obtained based on a cross-replicate comparison of the ASE ratio using eight $F_1$ hybrids, we confirmed this result through an estimation method using the average of ASE of all $F_1$ hybrids [16,40,41], which showed similar rates of *cis*- versus *trans*-regulation divergence (see Materials and methods, **S5 Fig**).

## *Cis*- and *trans*-regulatory effects on species-differential expression

We then examined whether the species-differentially expressed (SDE) genes in HVC and RA were affected by the transcriptional regulatory divergence between ZF and OF. Based on the RPKM values of each gene expressed in ZF and OF, 333 and 374 genes showed significantly different expression in HVC and RA, respectively, between the two species (2.9% and 3.3% of the total genes expressed in HVC and RA) (DEseq2 package, *p*-value corrected by the Benjamini-Hochberg method, $p < 0.05$; $n = 4$ each from ZF and OF) (**S6 Fig**). Totals of 209 and 242 genes of the SDE genes in HVC and RA, respectively, passed the ss-SNPs threshold for calculating the ASE ratio in $F_1$ hybrids. Such SDE genes were significantly enriched with a higher probability of *cis*-, *trans*-, and both *cis*- and *trans*-regulatory effects compared with those of non-SDE genes, in both HVC and RA (Fisher's exact test, ***$p < 0.001$) (**Fig 4C and 4D**). This shows a significant association of transcriptional regulatory changes with SDE genes in the song nuclei.

## A predominant effect on cellular molecular function by *trans*-regulatory divergence

To understand whether transcriptional regulatory divergence has any potential molecular contribution to cellular functions in HVC and RA, we performed GO enrichment analysis using the sets of genes affected by *cis*-, *trans*-, and both *cis*- and *trans*-regulatory changes. The result showed that more GO categories were enriched for *trans*-regulated genes compared with the other types of regulatory divergence in both HVC and RA (Fisher's exact test, *p*-value corrected by the Benjamini-Hochberg method) (**Fig 5A**). In particular, we found that GO

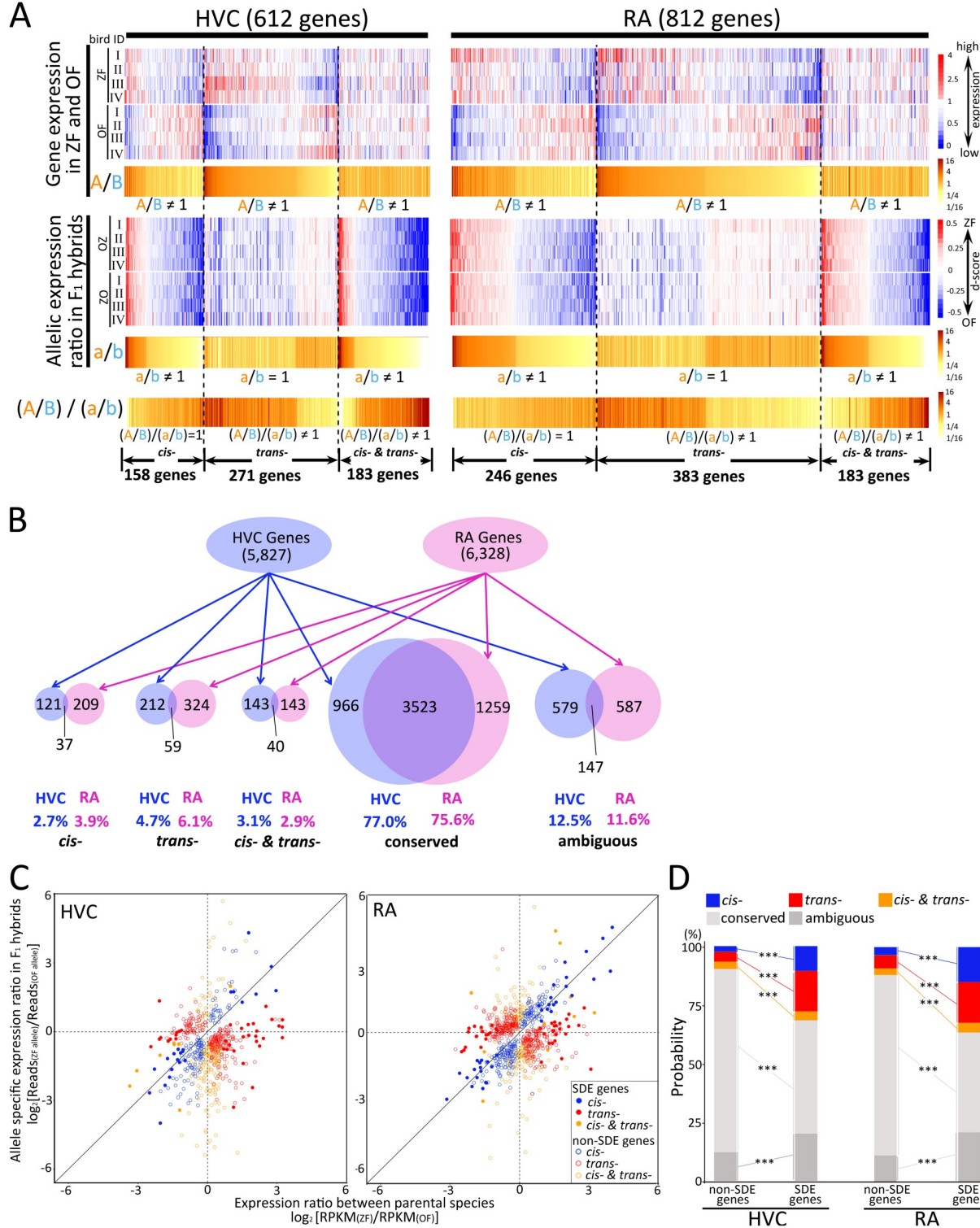

**Fig 4. Classification of transcriptional regulation divergence between ZF and OF.** (A) Heatmaps of gene expression in ZFs and OFs, and allelic expression ratios in F₁ hybrids for *cis-*, *trans-*, and both *cis-* and *trans-*regulated genes in song nuclei HVC and RA (blue–red colored). Comparison between species-different gene expression (A/B) and allelic expression ratios in F₁ hybrids (a/b) in heatmaps (dark brown–light yellow colored). "A" and "B" represent RPKM$_{(ZF\ average)}$ and RPKM$_{(OF\ average)}$, respectively. "a" and "b" represent Reads $_{(ZF\ allele)}$ and Reads $_{(OF\ allele)}$, respectively. (B) Gene numbers classified by *cis-*, *trans-*, both *cis-* and *trans-*, conserved, and ambiguous regulation in HVC and RA. (C) Scatterplots of expression ratios between ZF and OF (x-axis) and allelic expression ratios in F₁ hybrids (y-axis) for genes showing differential

expression between species. Blue-, red-, and orange-colored spots: *cis-*, *trans-*, both *cis-* and *trans-*regulated genes, respectively. Filled spots correspond to species-differentially expressed (SDE) genes. (D) *Cis-* and *trans-*effects on the expression of species-differentially regulated genes. The percent of *cis-*, *trans-*, both *cis-* and *trans-*, conserved, and ambiguous transcriptional regulatory genes in the SDE and non-SDE genes (Fisher's exact test, ***$p$ < 0.001). Relevant data values are included in S3 Data. F₁, first-generation; OF, owl finch; RA, robust nucleus of the arcopallium; RPKM, reads per kilobase of transcript per million reads mapped; ZF, zebra finch.

categories related to neural functions associated with presynapse, chemical synapse transmission, and neuron projection were significantly enriched for RA *trans-*regulated genes. These results motivated us to focus subsequently on altered *trans-*regulation in RA.

To predict the potential regulatory mediators driving species differences in the expression of *trans-*regulated genes in RA, we performed upstream regulatory analyses using Ingenuity Pathway Analysis (IPA) [42,43]. We found that brain-derived neurotrophic factor (BDNF) was the most significant upstream *trans-*mediator of genes under *trans-*regulation in RA, which included genes for neural plasticity and dendritic spine formation (glutamate decarboxylase [GAD] 2, NMDA glutamate receptor [GRIN] 2A, neuropeptide Y [NPY], and collapsin response mediator protein [CRMP] 1) (Fisher's exact test, $p = 6.44 \times 10^{-7}$) (Fig 5B and 5C and S7 Fig). Amino acid substitution and *trans-*mediator expression level changes could potentially mediate the *trans-*regulatory effects to alter downstream gene expression. In line with this prediction of BDNF as a *trans-*regulatory mediator in RA, we found two amino acid substitutions in BDNF between ZF and OF: Ser45Arg in prodomain and Thr143Met in nerve growth factor (NGF) domain (Fig 5D). Furthermore, BDNF was an SDE gene in HVC between the two species (Student's *t* test, *$p$ < 0.05) (Fig 5E). In HVC, as an upstream song nucleus connecting to RA, BDNF mRNA is primarily expressed in neurons projecting to RA [44], meaning that HVC could anterogradely secrete BDNF protein to RA via connecting axons as a potential *trans-*regulation via neural connections. Furthermore, we found differences between species regarding the regulation of the BDNF mRNA expression level in both HVC and RA: OFs had a higher expression level than ZFs at the 3-hour singing condition that induced singing-driven gene expression change, including BDNF (Fig 5F) [44,45]. Therefore, in order to uncover the putative *trans-*regulatory mechanisms of BDNF and to evaluate its potential impacts in generating species-specific songs, we examined how the amino acid substitution and/or expression level of BDNF relates to song structures.

## Correlation between individual variations of the species-biased song phenotypes and the BDNF expression level in F₁ hybrids

To evaluate the putative *trans-*regulatory effects mediated by the BDNF amino acid substitution or expression level, we investigated the correlation between song phenotypes and ASE ratio or the expression level of BDNF in F₁ hybrids. Considering that neither ZF nor OF are inbred, the interspecies F₁ hybrids might present individual variation in the ASE ratio and expression levels of transcribed genes. Consistently, at the transcriptome analysis in F₁ hybrids, we realized that F₁ hybrids possessed a wide range of individual difference in their ASE ratio and expression level of BDNF mRNA in HVC and RA (Fig 6A), such that each individual F₁ hybrid transcribed ZF- and OF-type BDNFs with a unique expression ratio and level. Furthermore, F₁ hybrids acquired individually unique songs with a wide range of ZF- and OF-biased features, even though they were reared listening to both ZF and OF songs as models (Fig 6B). We used the same sets of 7 total acoustic and sequential song parameters that showed differences between ZF and OF (5 for acoustic and 2 for sequential parameters) (Fig 2B and 2C). We found only one correlation between the ASE ratio of BDNF in RA and the entropy variance of syllables ($r = 0.800$, $p = 0.017$, Pearson correlation) (Fig 6C). In contrast, the expression

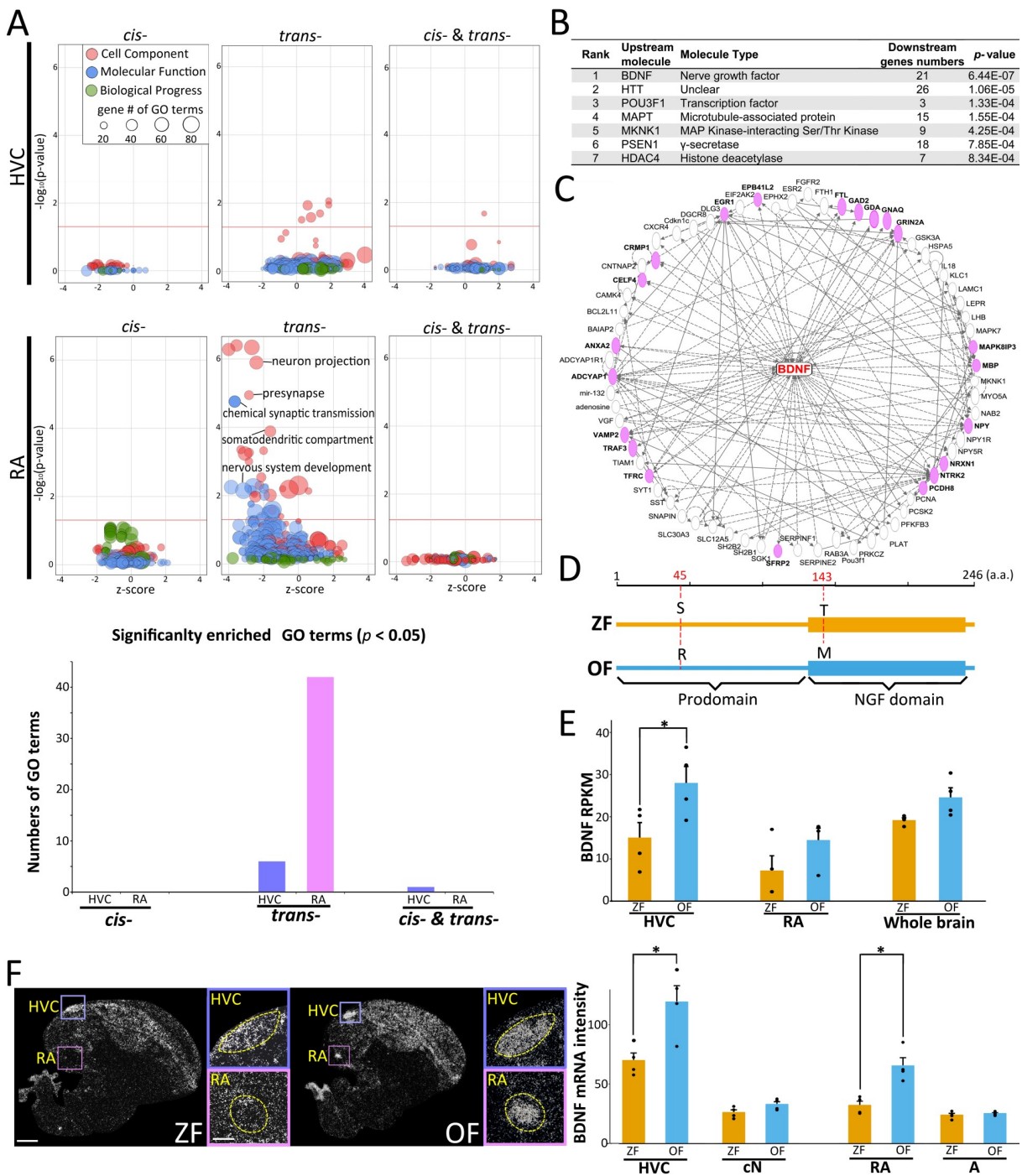

**Fig 5. Predominant effect on cellular molecular function by *trans*-regulatory divergence.** (A) GO enrichment analysis of the *cis*-, *trans*-, and both *cis*- and *trans*-regulated genes in HVC and RA. Size of points represents the number of genes assigned to each GO term. Red lines represent the *p*-value for significant enrichment (Fisher's exact test adjusted by the Benjamini-Hochberg method, *p* < 0.05). (B) Top 7 candidate upstream mediators for *trans*-regulated genes in RA. (C) Gene–gene connections for BDNF downstream genes. Pink-colored genes are *trans*-regulated genes in RA. Solid and dotted lines represent directed and undirected regulation, respectively, between connected genes. (D) Comparison of BDNF amino acid sequences between ZF and OF. (E) BDNF mRNA expression level in HVC, RA, and whole brain between ZF and OF at the silent condition based on RNA-seq data. **(F)** BDNF mRNA expression in the HVC, RA, and the surrounding areas (caudal nidopallium [cN] and archopallium [A], respectively) of ZF and OF at the 3-hour undirected singing condition (*n* = 4 each). White signals: BDNF mRNA. Scale bars, 1 mm (in left panes) and 200 μm (in right panel). Relevant data values are included in **S4 Data**. a.a., amino acid; BDNF, brain-derived neurotrophic factor; GO, Gene Ontology; NGF, nerve growth factor; OF, owl finch; RA, robust nucleus of the arcopallium; RNA-seq, RNA sequencing; RPKM, reads per kilobase of transcript per million reads mapped; ZF, zebra finch.

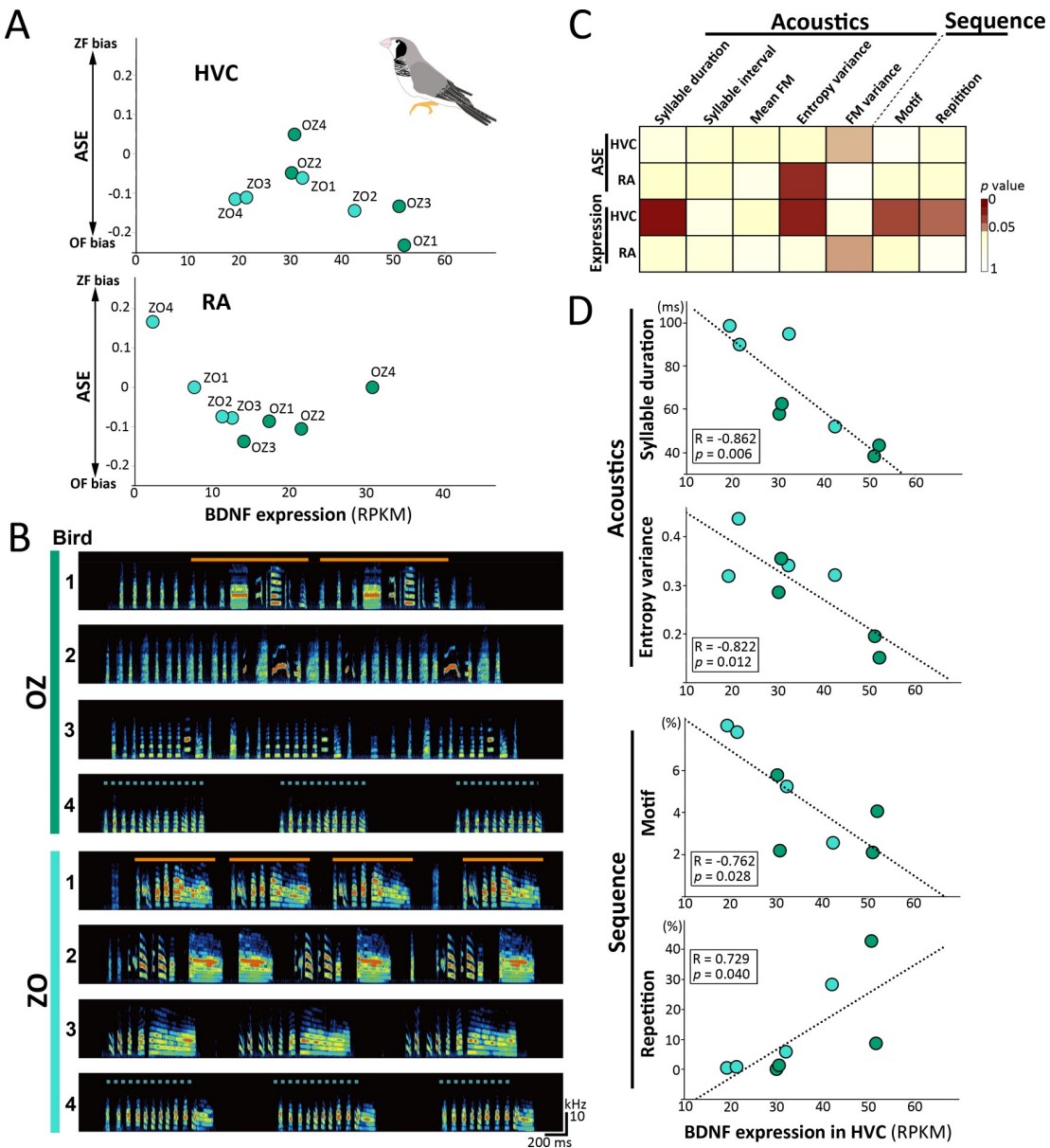

**Fig 6. Correlation between individual variation in BDNF expression level and species-biased song structures in $F_1$ hybrids.**
(A) Individual variation of BDNF mRNA expression level and ASE ratio between $F_1$ hybrids. **(B)** Individual variation of learned songs in $F_1$ hybrids that were tutored with ZF and OF songs. Orange solid and blue dotted lines represent the motif and repetitive structure of syllables, respectively. **(C)** Heatmaps showing the correlation of *p*-values between the BDNF expression level or ASE ratio and species-biased song phenotypes in $F_1$ hybrids. **(D)** Correlations between BDNF mRNA expression in HVC and species-biased song structures (syllable duration, entropy variance, motif, and repetition) among $F_1$ hybrid individuals. Relevant data values are included in **S5 Data**. ASE, allele-specific expression; BDNF, brain-derived neurotrophic factor; $F_1$, first-generation; OF, owl finch; OZ, $F_1$ hybrid offspring between OF♀ and ZF♂; RA, robust nucleus of the arcopallium; RPKM, reads per kilobase of transcript per million reads mapped; ZF, zebra finch; ZO, $F_1$ hybrid offspring between ZF♀ and OF♂.

level of BDNF mRNA in HVC had four significant correlations with acoustic and sequential song parameters in $F_1$ hybrids (acoustics: syllable duration [$r = -0.862$, $p = 0.006$] and entropy variance [$r = -0.822$, $p = 0.012$]; sequence: motif [$r = -0.762$, $p = 0.028$] and repetition [$r = 0.729$, $p = 0.040$], Pearson correlation) (**Fig 6C and 6D**). These correlational analyses in $F_1$ hybrids point to the BDNF mRNA expression level in HVC (instead of the amino acid

substitution) being the most likely RA *trans*-acting mechanism, which induces anterograde secretion of BDNA protein to RA.

### Alternation of *trans*-regulated gene expression and obliteration of learned song features by the pharmacological overactivation of BDNF receptors in the RA

To examine the potential causal links of transcriptional regulation between the BDNF concentration level and the predicted downstream *trans*-regulated genes, we infused a selective agonist of the BDNF receptor, i.e., tropomyosin receptor kinase B (TrkB), namely 7,8-dihydroxyflavone (7,8-DHF; 10 μg/μL) in vivo, into the RA of adult ZFs by using local retrodialysis (**S8 Fig**) [46]. Transcriptional analysis to compare control (PBS) and 7,8-DHF–infused birds revealed that 570 genes of the 11,655 genes expressed in the RA were differentially identified, with over 4-fold changes between the two groups (DEseq2, $p < 0.05$) (**Fig 7A**). Among the differentially expressed 570 genes, 6 of the 21 putative downstream *trans*-regulated genes of BDNF (shown in **Fig 5C**) were found to have an altered expression after the pharmacological activation of the BDNF receptors. This further supports our earlier finding that BDNF could be a potential regulatory mediator of the RA *trans*-regulated genes.

We also found that song changes following 7,8-DHF infusion, with a lower syllable transition consistency during the early stage (approximately 5 days after drug infusion). In addition, following continuous infusion for up to 2 weeks, adult structured songs gradually became more degraded, leading to the loss of learned song features in adult ZFs (**Fig 7B**). Although a few of the acoustic parameters (syllable duration and mean FM) maintained the original traits, syllable sequence (i.e., motif and repetitive indexes) and other acoustic parameters (i.e., intersyllable gap duration, entropy variance, and FM variance) were drastically changed by the infusion of 7,8-DHF (**Fig 7C and 7E**), thus indicating that a precise amount of BDNF contributes to the maintenance of the learned song structures of ZF.

## Discussion

Previous studies have demonstrated monogenic effects on adaptive behavioral phenotypes [7,47–49]. In contrast, the genetic basis of polygenic adaptations has been more challenging to pinpoint. Therefore, elucidating various SDE genes and the transcriptional regulatory divergences could be a promising step towards a better understanding of the contribution of multiple genes to the evolution of behaviors. For these, we examined the distribution of *cis*- and *trans*-regulatory divergences underlying the differences in gene expression in specific brain regions associated with the production of learned vocalizations between two closely related songbird species.

A number of studies that used entire organ tissues/body showed that there are more significant changes in *cis*- than *trans*-regulation between interspecies/lines of fruit flies [14,50], wasps [51], birds [52], and mouse [37]. In contrast, our study revealed that *trans*-regulatory changes were more prevalent than *cis*- in determining gene expression differences in song nuclei between two closely related species (**Fig 4B**). In addition, biological processes associated with neural functions were more enriched for genes showing *trans*-regulatory divergence in HVC and RA (**Fig 5A**). This difference in the effects of *cis*- or *trans*-regulations on transcriptional divergence could be caused by different methods of estimation using ASE ratio in the $F_1$ hybrids. However, even when we used an estimation method using the average ASE of $F_1$ hybrids, which has the potential to underestimate *trans*-regulation [40], we obtained a similar result showing that transcriptional regulatory divergence has occurred primarily in *trans*-regulation. To examine whether the *trans*-biased regulatory divergence is specific to song nuclei or

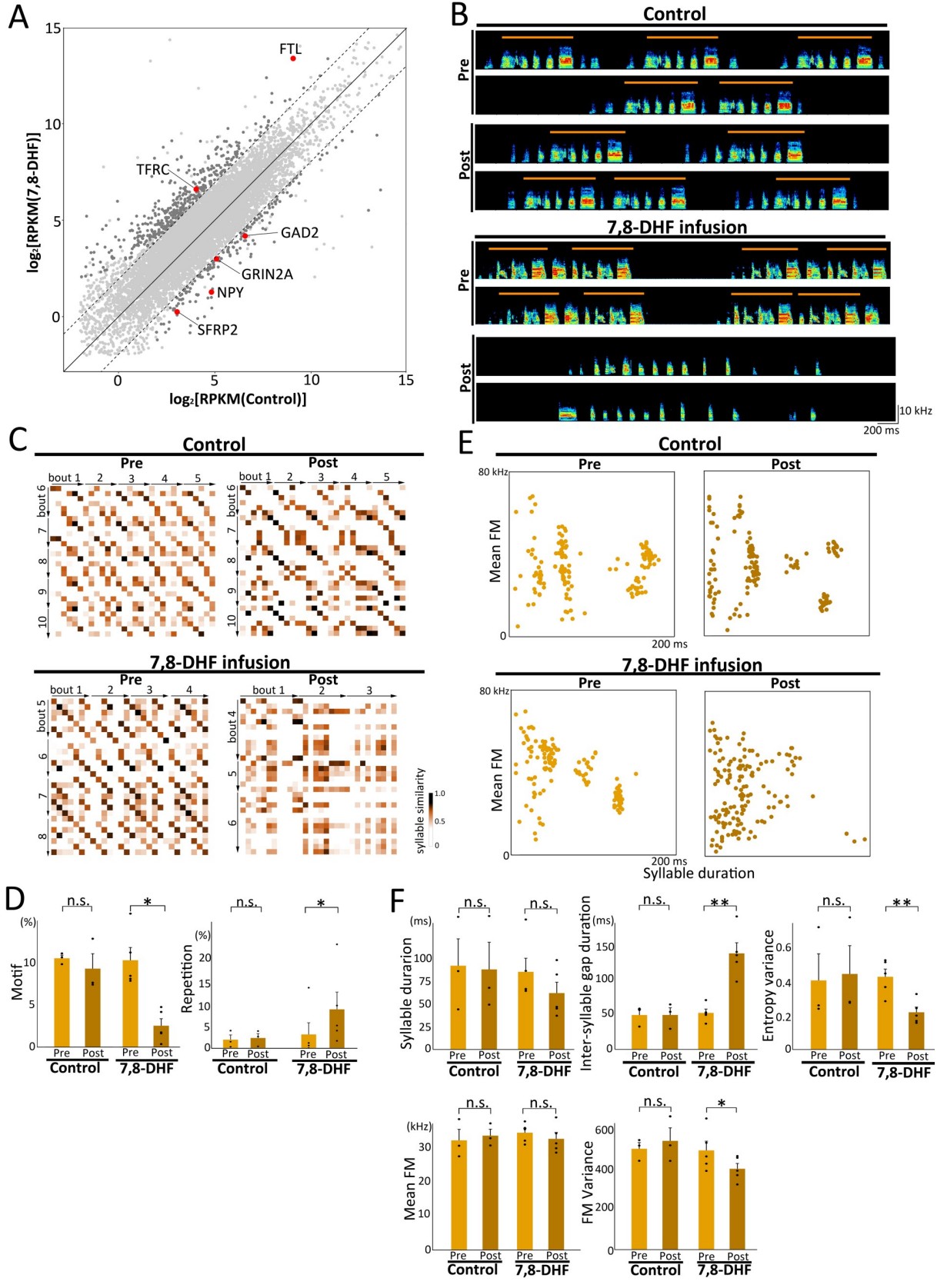

**Fig 7. Obliteration of species specificity of ZF song by BDNF receptor agonist infusion into RA. (A)** Scatterplot indicating RA gene expression in control and 7,8-DHF–infused birds. Dashed lines represent the boundary of the 4-fold expression difference. Darker gray colored dots represent significant differences in expressed genes higher than 4-fold between the control and 7,8-DHF–infused birds. Red colored dots represent downstream *trans*-regulated genes of BDNF (represented in Fig 5C). **(B)** Songs before and after infusing BDNF receptor TrkB agonist, 7,8-DHF. Typical examples of songs from control and 7,8-DHF–infused birds. Orange solid lines represent the motif structure of ZF songs. **(C)** Examples of syllable sequence changes between pre- and post-infusion. Syllable similarity matrices for a pair of songs produced by control and 7,8-DHF–infused birds. **(D)** Changes in the frequency of motif and repetition in songs at pre- and post-infusion stages (control ZF, $n = 3$, ZF with 7,8-DHF infusion [7–10 days], $n = 5$; paired $t$ test, $^*p < 0.05$). Each dot corresponds to individual birds. **(E)** Examples of syllable acoustic changes between pre- and post-infusion. Scatterplots indicate the distribution of 150 syllables (duration versus mean frequency) from control and 7,8-DHF–infused birds. **(F)** Changes in syllable acoustics (syllable duration, inter-syllable gap duration, entropy variance, mean FM, and FM variance) of songs at pre- and post-infusion stages (control ZF, $n = 3$, ZF with 7,8-DFH infusion [7–10 days], $n = 5$; paired $t$ test, $^{**}p < 0.01$, $^*p < 0.05$, n.s., not significant). Each dot corresponds to an individual bird. Relevant data values are included in **S6 Data**. BDNF, brain-derived neurotrophic factor; FM, frequency modulation; RA, robust nucleus of the arcopallium; RPKM, reads per kilobase of transcript per million reads mapped; TrkB, tropomyosin receptor kinase B; ZF, zebra finch; 7,8-DHF, 7,8-dihydroxyflavone.

not, it will be necessary to perform a similar analysis using samples from non-song nuclei or whole brain in songbirds. Furthermore, we set the cutoff with ≥5 reads at each ss-SNP position and median RPKM ≥ 10 to extract genes that were used for the calculation of the ASE ratio. This cutoff threshold is a stricter setting than other published studies [50,51]. Therefore, if we set a lower threshold to extract the ASE genes, the estimated gene number regulated by different transcriptional divergence would be increased.

In principle, two alleles in the cells of an $F_1$ hybrid are regulated in the same *trans*-regulatory environment. Therefore, differences in expression of two alleles in $F_1$ hybrids should reflect *cis*-regulatory divergence between the two parental genomes, generating a consistent ASE ratio among $F_1$ hybrids. However, at a considerable number of genes in the song nuclei, we observed a large variance in ASE ratio among $F_1$ hybrids, which we defined as "ambiguously" regulated genes (**S2 Fig**). Approximately 10% of the total expressed genes in HVC and RA were categorized as "ambiguous" (**Fig 4B**). Ambiguous regulation could result from intraspecies genomic variation. Indeed, the experimental ZF and OF have not been genetically selected animals. We found 742,302 and 414,040 polymorphic SNPs in the transcribed sequence from the whole brain of ZFs and OFs, respectively ($n = 4$ each). Therefore, the individual variability in the ASE ratio between $F_1$ hybrids may be caused by intraspecies polymorphisms, which could in turn be additional *trans*- and *cis*-regulatory variants underlying individual difference in gene expression in song nuclei.

We found that BDNF is one of several potential upstream mediators for *trans*-regulated genes in RA (**Fig 5 and S7 Fig**). BDNF transcription, secretion, and actions are directly regulated by neural activity. Secreted BDNF mediates multiple activity-dependent processes, including neuronal differentiation/growth, synapse formation, and plasticity during developmental and adult stages [53–56]. In the song system of songbirds, singing behavior induces BDNF mRNA expression in song nuclei including HVC, suggesting that neural activity-dependent signaling of BDNF regulates neuronal maturation [44,45,57,58]. We had reported that ZFs prevented from singing during the song learning period possess immature dendritic spine density in RA neurons and produced highly unstable song lacking species-specific features when allowed to sing freely, even at the adult stage [59]. Although transient BDNF up-regulation in HVC enhances song learning during the critical period [60], a short-term local injection of BDNF into RA of adult ZFs changed crystallized songs to juvenile-like plastic songs with sequence variability; these changes correlated with an increase in HVC axonal boutons in RA [61]. We further confirmed that the continuous and local infusion of BDNF receptor agonist 7,8-DHF into RA of adult ZFs induced severe song degradation, eliminating both learned acoustic and sequence features. Therefore, we suggest that BDNF mediates the precise synaptic connections and strength of connections allowing HVC to activate populations of RA

neurons at specific and precise time points during song rendition. Species differences in BDNF expression in the song nuclei could shape the anatomical and molecular bases for the generation of species-specific learned song structure via the activity-dependent *trans*-regulation of the downstream genes. Using $F_1$ hybrids, we found that ASE ratios of BDNF were more weakly associated with species-biased song phenotypes compared with BDNF mRNA expression levels. However, this does not rule out a potential *trans*-regulatory effect mediated by BDNF amino acid substitution on species-specific behaviors. In human, the BDNF polymorphism (Val66Met; rs6265) affects intracellular trafficking and reduces activity-dependent secretion of mature BDNF [62]. The BDNF polymorphism causes altered dendritic spine density, memory formation, and extinction [63,64]. Future application of genome editing technologies would be a powerful tool to elucidate the in vivo contribution of BDNF polymorphisms to species-specific behaviors.

We investigated the divergence between the ZF and OF in terms of gene transcription for the generation of species-specific learned songs (but not for the learning process). Thus, we performed a series of experiments including song comparative analysis, comprehensive RNA sequencing (RNA-seq), and BDNF agonist infusion by using adult birds after the critical period for song learning. However, it is crucial to consider the potential effects of BDNF on the development of neural circuits for species-specific song learning and production during the embryonic and early post-hatching periods. Although we observed that the pharmacological overactivation of BDNF receptors drastically affected song change and led to the loss of learned song structures at both syllable acoustic and sequence levels, we cannot tell whether such song degradation is induced by any species-specific deficiency. Given the wide variety of BDNF cellular functions, the pharmacological experiment was limited by the selective modification of signaling machinery for species-specific song generation. We found that not only predicted downstream *trans*-regulated genes but also over 550 genes had altered expression levels in the RA, as assessed by comparing control and 7,8-DHF–infused birds. Therefore, future research with more refined experiments for targeted multiple genes, manipulation timing, and cell types will be crucial.

In this study, we investigated the regulatory drivers of species divergence in gene expression in song nuclei in the vocal motor circuit in adults. We suggest these regulatory differences between species could explain a genetic molecular mechanism for the generation and maintenance of the species specificity of learned songs. The anterior forebrain pathway (AFP) is a cortico-basal ganglia-thalamocortical loop, which is a specific pathway for song learning during development and for vocal plasticity maintenance later in life [65–69]. For sensorimotor coordination, AFP generates instructive biased variability and conveys this to the premotor song nuclei RA as a reinforcement signal [46,70]. Currently, we cannot make direct causal links between AFP function and the acquisition of species-specific song patterns. However, lesion of the basal ganglia nucleus, Area X of the striatum (Area X), in the AFP at an early critical period was shown to disrupt motif structure, a sequential trait commonly observed in ZF songs [69]. Furthermore, the expression of transcription factors such as Forkhead box protein P2 (FoxP2) and androgen receptors in Area X shows species-specific patterns [25,26]. These transcription factors could be potential regulators for further species difference due to their regulatory effects on downstream genes, thereby generating species-biased vocal plasticity, which in turn promotes species-specific song learning. Therefore, studying species differences in gene expression in the song nuclei of the AFP through the critical period of song learning would provide vital insight into how species-specific patterns of gene expression underlie species-specific songs.

In conclusion, our results suggest a neurogenetic association between brain region–specific transcriptional divergence and species-specific learned behaviors. Most complex motor skills,

such as birdsong and human speech, are acquired through learning and constrained in a species-specific manner [35,71]. Using similar techniques to those developed in the present study on other interspecific hybrids could give additional insights into the existence of more conserved or unique *cis-/trans*-regulatory changes underlying the evolution of species-specific learned behaviors.

## Materials and methods

### Ethics statement

All experiments were conducted under the guidelines and approval of the Committee on Animal Experiments of Hokkaido University (Approved No. 18–0053). These guidelines are based on the national regulations for animal welfare in Japan (Law for the Humane Treatment and Management of Animals with partial amendment No. 105, 2011). For brain sampling, the birds were humanely killed by decapitation after injection of an overdose of pentobarbital.

### Animals and song tutoring

ZFs (*T. guttata*) and OFs (*T. bichenovii*) were obtained from our breeding colony at Hokkaido University and local breeders. Reciprocal $F_1$ hybrids were bred by pairing ZF and OF at our breeding colony. All birds were maintained with food and water available ad libitum under a 13:11-hour light/dark cycle. For song cross-tutoring experiments, ZF chicks were raised by both parents in breeding cages until 10–15 phd, and then the father was removed by 15–25 phd from the cage to prevent male juveniles from listening their father's song. OF chicks were hand-raised after hatching until they could feed themselves (approximately phd 30–40). After fledging, juveniles were subsequently housed in individual isolation boxes and then individually housed in a sound-attenuating box containing a mirror to reduce social isolation. Cross-species' tutor songs were played 7 times each in the morning and afternoon at 55–75 decibels from a speaker (SRS-M30, SONY, Tokyo, Japan) passively controlled by Sound Analysis Pro. Similarly, $F_1$ hybrids were song tutored by passively and randomly playing a set of ZF and OF songs with an interval duration of 300–500 ms as the song model.

### Song recording and analysis

Songs were recorded using a unidirectional microphone (SM57, Shure) connected to a computer with Sound Analysis Pro (SAP v1.04). For analysis of the acoustic features of songs, 500 syllables were randomly selected from ZF and OF songs (*n* = 6 birds each). To characterize the syllable that differed between ZF and OF, a total of 10 acoustic features were measured: syllable duration, inter-syllable gap duration, mean pitch, pitch goodness, Wiener entropy, entropy variance, mean AM, AM variance, mean FM, and FM variance [31]. Statistical analysis was performed on these acoustic features between ZF and OF by one-way ANOVA. For the analysis of the sequence feature of songs (motif and repetition rates in a song), a syllable similarity matrix (SSM) analysis was performed following a previously reported method [30] (**S1 Fig**). This method calculates the contiguous syllables transition frequency of "paired (motif)" and "repetitive" syllables transitions in the songs. To test song structure changes by pharmacological manipulation, we measured both the syllable acoustic and sequential parameters of 150 syllables at pre- and post-time points (7–10 days) after drug infusion. Six acoustic syllable parameters (syllable duration, inter-syllable gap duration, entropy variance, AM variance, mean FM, and FM variance) and sequence features (motif and repetition transition rates) were used for the PCA and 2D view, and this was performed using the prcomp and rgl packages in R, respectively.

## Brain tissue sampling and RNA extraction for RNA-seq

For sampling of whole brain tissues, adult male birds were isolated in a soundproof chamber for at least one day before humanely killing (ZF: $n = 4$, OF: $n = 4$, adult [>200 phd]). Birds were killed under silent and dark condition in the morning before the lights were turned on. The pallium and pallidum regions were rapidly dissected, frozen on dry ice, and stored at −80°C until RNA extraction. Total RNA was isolated using TRIzol Reagent according to the manufacture's protocol (Invitrogen) and was treated with RNase-free DNase.

For sampling of HVC and RA tissues by laser capture microdissection (LCM), adult ZF, OF, ZO, and OZ $F_1$ hybrids ($n = 4$ birds each, >130 phd) and control and 7, 8-DHF–infused ZFs ($n = 3$ each, >130 phd) were isolated in sound-attenuation boxes and killed under silent and dark condition. Brains were embedded in OCT compound (Sakura Fine Technical) and stored at −80°C until use. Brain sections were cut at a 14-μm thickness in the sagittal plane and mounted onto glass slides with a handmade membrane system for laser microdissection. We confirmed the presence and boundaries of HVC and RA using Nissl staining (LCM Staining kit; Ambion). HVC and RA were microdissected using a laser capture microscope ArcturusXT (Arcturus Bioscience) with the following parameter settings: spot diameter, 100 μm; laser power, 80 mW; and laser duration, 80 ms [72]. The captured tissues were dissolved into RLT buffer (Qiagen) with $\beta$-mercaptoethanol, treated with DNase in the column to avoid contamination of genomic DNA, and then stored at −80°C until RNA extraction.

## RNA-seq library construction and sequencing

RNA integrity number (RIN) and concentration were measured with Bioanalyzer 2100 (Agilent Technologies) to guarantee the quality of RNA. For RNA-seq of HVC and RA, we performed first-strand cDNA amplification using total RNA (1–2 ng) from HVC and RA under a PCR amplification condition of 14 cycles at 98°C for 10 seconds, 65°C for 15 seconds, and 68°C for 5 minutes, following the Quartz-amplification method [73]. Amplified cDNAs were purified using a PCR purification column (MiniElute PCR Purification Kit; Qiagen) and the concentration was measured using Bioanalyzer 2100 (Agilent Technologies). Non-amplified first-strand cDNAs synthesized using total RNA from the whole brain (telencephalon) and amplified cDNAs using total RNA from HVC and RA tissue were used to construct poly(A) selected paired-end sequencing libraries (TruSeq DNA Sample Prep Kits, Illumina). All libraries were sequenced using the Illumina Hiseq2500 platform for 100-bp paired-end sequencing.

For each telencephalon brain sample, 33.5–47.0 M RNA-seq reads were output from the Illumina Hiseq 2500. Sequencing reads were mapped onto the ZF reference genome obtained from Ensembl (*Taeniopygia_guttata* taeGut3.2.4.dna.fa) with the Tophat2 program and assembled to predicted transcripts with the Cufflinks program. Through comparison with the previous annotation file using the cuffcompare program, 12,156 transcripts were identified as predicted RNA transcripts expressed in the ZF telencephalon. All RNA-seq data were deposited in the DDBJ Sequence Read Archive (submission numbers DRA005548, DRA002970, and DRA008696).

## Identification of ss-SNPs

Adapter sequences of raw data from ZF and OF whole brain NGS results were removed by Trimmonatic. Clean reads from ZF and OF whole brain were mapped to a ZF reference genome obtained from Ensembl (*Taeniopygia_guttata*.taeGut3.2.4) by TopHat2 to reconstruct pseudo ZF and OF genomes. Mapped reads with longer gaps (>3,000 bp) were removed in the subsequent analysis. ss-SNPs and insert and deletions (indels) between ZF and OF were identified from the mapping result of the whole brain reads. The positions of ss-SNPs and indels

were used to reconstruct pseudo genomes of ZF and OF. ss-SNPs were defined as follows: the base variants were same in all individuals of a species, but different from the base found in all individuals of another species. SNPs in individuals of the same species (intraspecies SNPs) were maintained as the same base for both ZF and OF reconstructed genome sequences. MUMmer software was used to identify ss-SNPs using the reconstructed ZF and OF genomes.

## Read mapping and quantification of gene expression level

Low-quality reads and adaptor sequences were removed from all HVC and RA raw reads using the Filter FASTQ pipeline (https://cell-innovation.nig.ac.jp) and Flexbar software. Clear reads were mapped to reconstructed ZF genome by TopHat2. Transcript levels were quantified as RPKM value. Cufflinks was used to evaluate the expression levels of each gene by calculating the RPKM of HVC and RA samples of ZF and OF using the improved genome annotation Gene Transfer Format (GTF) file [59]. Based on the RPKM of individual birds ($n = 4$ each from ZF and OF; $n = 3$ each from 7,8-DHF–infused and control ZFs), the expression differences of each gene were identified between ZF and OF and between 7,8-DHF and control ZFs as differently expressed genes using the R package DEseq2 (adjusted $p$-value $< 0.05$, the Benjamini-Hochberg procedure).

## Allelic expression ratio in $F_1$ hybrids

To distinguish reads of the two alleles in $F_1$ hybrids, the mapping results of HVC and RA of $F_1$ hybrids were used following SNPsplit's instruction. First, an N-marked genome sequence was constructed by replacing "N" at the ss-SNP position in the ZF pseudo genome. RNA-seq reads of HVC and RA of ZF, OF, and $F_1$ hybrids were mapped to the N-marked genome by TopHat2. ss-SNPs were identified as SNP sites having more than 98% of total reads that were different between ZF and OF alleles. The identified ss-SNPs were reverified by reads from HVC and RA of ZF and OF, to enhance the reliability of ss-SNPs. The mapped HVC and RA reads of $F_1$ hybrids were then separated into ZF or OF allele transcripts based on the ss-SNP information, and the number of reads was counted at each ss-SNP position by SAMtools.

The following thresholds were set for calculating the allelic expression ratio of each gene expressed in HVC and RA of $F_1$ hybrids: (i) existence of at least one ss-SNP, (ii) more than 5 reads at each ss-SNP site, and (iii) a median RPKM of at least 10 for all 16 individuals (including ZF, $n = 4$; OF, $n = 4$; ZO, $n = 4$; OZ, $n = 4$). The allelic expression ratio was quantified using the $d$-score [74]:

$$d = \frac{\text{Reads}_{(ZF)}}{\text{Reads}_{(ZF)} + \text{Reads}_{(OF)}} - 0.5$$

$d$-scores of 0 reflect equal expression between the two alleles, whereas $d$-scores of $-0.5$ and $0.5$ reflect exclusive transcription from OF or ZF alleles, respectively.

## Identification of *cis*- and/or *trans*-regulatory divergence

The potential of genomic imprinting in $F_1$ hybrids was tested using Spearman's rank correlation of gene allelic expression ratio between ZO ($n = 4$) and OZ ($n = 4$). The difference in the allelic expression ratio of each gene was compared between ZO and OZ hybrids using one-way ANOVA (ZO, $n = 4$; OZ, $n = 4$; adjusted $p$-value by the Benjamini-Hochberg method).

*Cis*- and/or *trans*-regulatory divergences were evaluated using a previously reported method [16]. The gene expression ratio between parental species was calculated with the formula $X = \log_2(A/B)$, where "X" is the gene expression ratio between parental species; "A" and

"B" are the average RPKM for ZFs and OFs, respectively ($n = 4$ each from ZF and OF). The allelic expression ratio of $F_1$ hybrids was calculated as $Y = \log_2(a/b)$, where "Y" is the allelic expression ratio between two alleles; "a" and "b" are the read counts of ZF and OF alleles in $F_1$ individuals, respectively. *Cis*- and *trans*-effects on gene expression divergence were estimated by the scheme described in **Fig 1D**. In brief, the regulation mechanism of gene expression between ZF and OF was (1) a *cis*-regulatory difference if X = Y and Y $\neq$ 0; (2) a *trans*-regulatory difference if X $\neq$ Y and Y = 0; (3) both *cis*- and *trans*-regulatory differences if X $\neq$ Y and Y $\neq$ 0; (4) no *cis*- and *trans*-regulatory differences (i.e., conserved) if X = Y and Y = 0. The Student's *t* test was used to determine the difference between the gene expression ratio in parental species and the allelic expression ratio in $F_1$ hybrids. The SGoF program was employed to correct *p*-values for multiple testing (adjusted $p \leq 0.05$). The previous standard method for estimating regulatory divergence can lead to a negative correlation as an artifact when *cis*-estimates have any errors [40, 41]. To avoid this bias, first we randomly selected four individual $F_1$ hybrids as a group to estimate *cis*-effects using their average ASE ratio while the remaining four $F_1$ individual hybrids were used to compare the expression ratio between ZF and OF. For each gene, a total of 70 combinations were constructed by random selection of four of eight $F_1$ hybrid birds ($n = 4$ each from ZO and OZ). Thus, *cis*- and/or *trans*-regulatory identification was done for each gene for each pair of 70 total combinations. During this cross-replicate comparison, some genes were categorized as different transcriptional regulations due to a large variance in ASE ratios among $F_1$ individuals. Therefore, we finally determined which transcriptional divergence made the main regulatory effect on each gene by two steps of statistics following (i) calculation of the difference between four categories (*cis*-, *trans*-, both *cis*- and *trans*-, and conserved) using the Chi-squared test (with adjusted *p*-value by FDR < 0.05) and (ii) a comparison of the difference between the first- and second-strongest regulatory effects using a Fisher's exact-test (adjusted *p*-value by FDR < 0.05). If genes did not show significance at both tests, such genes were defined as "ambiguous regulatory genes" (**S2 Fig**).

In addition, we performed analysis of *cis*- and/or *trans*-regulatory divergence using a standard method [37,51] and compared these results with those from the above method. The difference of the standard method is that the allelic expression ratios of all eight $F_1$ hybrids (ZO = 4, OZ = 4) were used to estimate *cis*- and *trans*-regulatory effects. In brief, the parental expression ratio value X and the allelic expression ratio in $F_1$ hybrid value Y were calculated similarly to our new method. The average values Y of eight $F_1$ hybrid individuals were compared with values X and 0, respectively, to estimate *cis*- and *trans*-effects by the scheme described in Fig 1D (Student's *t* test). The SGoF program was employed to perform multiple testing correction (adjusted *p*-value $\leq$ 0.05) (**S5 Fig**).

### Functional analysis of *cis*- and/or *trans*-regulated genes

The functions of genes with *cis*-, *trans*-, and *cis*- and *trans*-regulatory divergences between ZF and OF in HVC and RA were annotated by GO analysis (DAVID Bioinformatics Resources 6.8; https://david.ncifcrf.gov). GO enrichment analysis was performed for each gene group using Fisher's exact tests (*p*-value was adjusted by the Benjamini-Hochberg method). As *trans*-regulated genes in RA were enriched for the most GO terms, an upstream regulatory analysis was performed for RA *trans*-regulated genes using IPA software.

### In situ hybridization

BDNF cDNA fragments used for the synthesis of in situ hybridization probes were cloned from a whole-brain cDNA mixture of a male ZF. Total RNA was transcribed to cDNA using Superscript Reverse Transcriptase (Invitrogen) with oligo dT primers. The cDNAs were

amplified by PCR using oligo DNA primers directed to the open reading frame region from the NCBI cDNA database. PCR products were ligated into the pGEM-T Easy plasmid (Promega). The cloned sequences were searched using NCBI BLAST/BLASTX to compare with homologous genes to other species and genome loci identified using BLAT of the UCSC Genome Browser.

Adult male ZFs ($n = 4$) and OFs ($n = 4$) were used. Birds were individually housed in sound-attenuating boxes overnight. On the following morning, singing behavior (undirected singing) was recorded for 3 hours after the lights were turned on. After each singing behavior observation session, the birds were euthanized by decapitation. Brains were embedded in OCT compound (Sakura Fine Technical) and stored at −80˚C until use. Frozen sections (12-μm thick) were cut in the sagittal plane. Brain sections for a given experiment were simultaneously fixed in 3% paraformaldehyde/1× PBS (pH 7.4), washed in 1× PBS, acetylated, dehydrated in an ascending ethanol series, air-dried, and processed for in situ hybridization with antisense $^{35}$S-UTP–labeled riboprobes of genes. To generate the riboprobes, gene inserts in the pGEM-T Easy vector were PCR amplified with plasmid M13 forward and reverse primers and then gel purified. The amplified DNA fragments and SP6 or T7 RNA polymerase were used to transcribe the antisense $^{35}$S-riboprobes. A total of $1 \times 10^6$ cpm of the $^{35}$S-probe was added to a hybridization solution (50% formamide, 10% dextran, 1× Denhardt's solution, 12 mM EDTA [pH 8.0], 10 mM Tris-HCl [pH 8.0], 300 mM NaCl, 0.5 mg/mL yeast tRNA, and 10 mM dithiothreitol). Hybridization was performed at 65˚C for 12–14 hours. The slides were washed in 2× SSPE and 0.1% $\beta$-mercaptoethanol at room temperature for 1 hour; 2× SSPE, 50% formamide, and 0.1% $\beta$-mercaptoethanol at 65˚C for 1 hour; and 0.1× SSPE twice at 65˚C for 30 minutes each. Slides were dehydrated in an ascending ethanol series and exposed to X-ray film (Biomax MR, Kodak) for 1–14 days. We carefully attended in order not to overexpose X-ray films to S$^{35}$-riboprobe hybridized brain sections. The slides were then dipped in an autoradiographic emulsion (NTB2, Kodak), incubated for 1–8 weeks, and processed with D-19 developer (Kodak) and fixer (Kodak). For quantification of mRNA signal, exposed X-ray films of brain images were digitally scanned under a microscope (Leica, Z16 APO) connected to a CCD camera (Leica, DFC490) with Application Suite V3 imaging software (Leica), as previously described [45, 72, 75, 76]. To minimize handling bias for signal detection among experimental groups, we performed in situ hybridization using multiple brain sections at once for each probe and exposed S$^{35}$-riboprobe hybridized brain sections on the same sheet of X-ray films. The same light settings were used for all images. Photoshop (Adobe Systems) was used to measure the mean pixel intensities in the brain areas of interest from sections after conversion to 256 grayscale images.

## Pharmacological manipulation

Custom microdialysis probes were built using a microdialysis membrane (SpectralPor, in vivo microdialysis hollow fiber, O.D. = 216 μm; total weight, <0.035 g) attached to a drug reservoir, based on a previously described method [46]. Probes were bilaterally implanted at positions adjacent to RA using stereotaxic coordinates. Before setting the probe, spontaneous neural activity was measured to verify the location of RA. Microdialysis probes were carefully set outside the RA to avoid physically damaging the RA, because damage to the RA could induce song changes. Following surgery, the reservoir was filled every morning with saline until the bird began to sing consistently and its phonological and syntactical features were confirmed not to be damaged by implantation of probe. To ensure the position of microdialysis probes, tetrodotoxin (TTX; 6–12 μM) was infused into the RA in a hemisphere, and a hemi-RA inactivation-induced song change was confirmed. Saline ($n = 3$ birds) or 7,8-DHF (10 μg/μL in 0.9%

NaCl, pH 7.4–7.6, Santa Cruz; $n = 5$ birds) was then continuously infused during daytime via the injection of approximately 2.5 uL of solution into the outer reservoir of the microdialysis probes 3 to 4 times daily. The manipulated birds were allowed to move freely in a sound-attenuation chamber, and the song of each individual was recorded over 10 days after initiation of drug infusion. The remaining drug volume and infusion speed were checked by using a transparent polyimide tube as the outer reservoir of the microdialysis probes. Probe positioning was evaluated postmortem by histological staining of tissue sections.

## Supporting information

**S1 Fig. ZF and OF species-specific song features.** (A) (Upper panels) SSM analysis for the detection of syllable sequential transition patterns. The SSM comprises two steps: First, a correlation matrix including the syllable similarity scores was prepared using the round-robin comparison of all syllables in two songs to maintain the sequential order of the syllables in the songs. These similarity scores in the matrix were binarized at a threshold at 0.595. Second, the occurrence rate of two patterns of binarized "2 row × 2 column" cells in the SSM was calculated as a percentage of the paired (motif) and repetitive-syllable transition types (see the Materials and methods). (Lower panel) Test examples of the SSM method using artificial song models mimicking the songs with motif and repetitive sequences. (B) The similar distribution range of syllable acoustic traits between ZF and OF. Violin plots of the distribution of syllable duration, inter-syllable gap duration, entropy variance, AM variance, mean FM, and FM variance from ZF and OF that were reared with conspecific song tutoring (total 3,000 syllables from $n = 6$ birds each and 500 syllables/bird). (C) PCA of the song features of ZFs and OFs reared under conspecific and cross-species song tutoring conditions ("Con": $n = 6$ each from conspecific song tutored ZF and OF; "Cross": $n = 4$ and 3 from cross-species song tutored ZF and OF, respectively). Relevant data values are included in **S1 Data** for panels **B and C**. AM, amplitude modulation; FM, frequency modulation; OF, owl finch; PCA, principal component analysis; SSM, syllable similarity matrix; ZF, zebra finch.
(TIF)

**S2 Fig. Experimental flowchart for the calculation of species-differently expressed genes and characterization of transcriptional regulatory divergence.**
(TIF)

**S3 Fig. Species differences in gene expression in HVC between ZF and OF.** (Left panels) Expression levels of PRKAA1, NR2E1, and CACNA1E in song nucleus HVC of ZFs and OFs. Gray-colored boxes represent the position of exons for each gene. Dark blue peaks below exons represent read density. (Right panels) Gene expression levels in ZF and OF. Each dot represents RPKM value for the individual. Mean ± SEM ($n = 4$ birds each, one-way ANOVA, $^*p < 0.05$; n.s., not significant). Relevant data values are included in **S2 Data**. CACNA1E, calcium voltage-gated channel subunit alpha 1E; NR2E1, nuclear receptor subfamily 2 group E member 1; OF, owl finch; PRKAA1, protein kinase AMP-activated catalytic subunit alpha 1; RPKM, reads per kilobase of transcript per million reads mapped; ZF, zebra finch.
(TIF)

**S4 Fig. No genomic imprinting genes in reciprocal $F_1$ hybrids of ZF and OF.** Scatterplots of allelic expression ratios of 5,849 and 6,328 genes in HVC and RA, respectively, of OZ and ZO hybrids (Spearman correlation coefficient). Relevant data values are included in **S3 Data**. $F_1$, first-generation; OF, owl finch; OZ, $F_1$ hybrid offspring between OF♀ and ZF♂; RA, robust nucleus of the arcopallium; ZF, zebra finch; ZO, $F_1$ hybrid offspring between ZF♀ and OF♂.
(TIF)

**S5 Fig. *Cis-*, *trans-*, both *cis-* and *trans-*, and conserved regulation in HVC and RA estimated by a method using the average of ASE of all F$_1$ hybrids.** Relevant data values are included in S3 Data. ASE, allele-specific expression; F$_1$, first-generation; RA, robust nucleus of the arcopallium.
(TIF)

**S6 Fig. SDE genes in HVC and RA.** (A) SDE genes in HVC and RA. Orange- and blue-colored spots represent significantly higher expression in ZF or OF, respectively (DEseq2 corrected with the Benjamini-Hochberg method, $p < 0.05$). (B) Venn diagram representing the number of genes in HVC and RA that are differently expressed between ZF or OF. Relevant data values are included in S3 Data. OF, owl finch; RA, robust nucleus of the arcopallium; SDE, species-differentially expressed; ZF, zebra finch.
(TIF)

**S7 Fig. Gene–gene connections driven by the top 7 candidate upstream mediators for *trans*-regulated genes in RA.** Top 7 candidate upstream mediators, including BDNF, HTT, POU3F1, MAPT, MNKK1, PSEN1, and HDAC4. *Trans*-regulated genes by BDNF in RA are noted in red. Orange- and green-colored genes are *trans*-regulated genes that are significantly expressed more highly in RA of ZF or OF, respectively. Relevant data values are included in S4 Data. BDNF, brain-derived neurotrophic factor; HDAC4, histone deacetylase 4; HTT, huntingtin; MAPT, microtubule-associated protein tau; MNKK1, MAP kinase-interacting serine/ threonine protein kinase 1; OF, owl finch; POU3F1, POU class 3 homeobox 1; PSEN1, presenilin 1; RA, robust nucleus of the arcopallium; ZF, zebra finch.
(TIF)

**S8 Fig. Untethered microdialysis for pharmacological manipulation of BDNF receptors in RA.** (Left) Photograph of homemade microdialysis probe. (Right) A ZF with microdialysis probes bilaterally implanted in RA. BDNF, brain-derived neurotrophic factor; RA, robust nucleus of the arcopallium; ZF, zebra finch.
(TIF)

**S1 Data. Underlying data for Fig 2B and 2C and S1 Fig.**
(XLSX)

**S2 Data. Underlying data for Figs 3A and 2B and S3 Fig.**
(XLSX)

**S3 Data. Underlying data for Fig 4A–4D and S4–S6 Figs.**
(XLSX)

**S4 Data. Underlying data for Fig 5A, 5B, 5E and 5F and S7 Fig.**
(XLSX)

**S5 Data. Underlying data for Fig 6A, 6C and 6D.**
(XLSX)

**S6 Data. Underlying data for Fig 7A and 7C–7F.**
(XLSX)

## Acknowledgments

We thank Keiko Sumida for her excellent bird care and breeding, Drs. C. N. Asogwa and D. Wheatcroft for their comments and discussion, Drs. M. Tanaka and K. Hamaguchi for their

instruction for making custom microdialysis probes, and Dr. Y. Suzuki's laboratory in the Department of Computational Biology, the University of Tokyo, for RNA-seq experiments.

## Author Contributions

**Conceptualization:** Hongdi Wang, Kazuhiro Wada.

**Data curation:** Hongdi Wang, Azusa Sawai, Noriyuki Toji, Shin Hayase, Satoru Akama, Jun Sese, Kazuhiro Wada.

**Formal analysis:** Hongdi Wang, Satoru Akama, Jun Sese, Kazuhiro Wada.

**Funding acquisition:** Hongdi Wang, Kazuhiro Wada.

**Investigation:** Hongdi Wang, Azusa Sawai, Noriyuki Toji, Kazuhiro Wada.

**Methodology:** Hongdi Wang, Azusa Sawai, Noriyuki Toji, Rintaro Sugioka, Yu Ji, Shin Hayase, Satoru Akama, Jun Sese, Kazuhiro Wada.

**Project administration:** Kazuhiro Wada.

**Resources:** Hongdi Wang, Azusa Sawai, Yukino Shibata, Yuika Suzuki, Kazuhiro Wada.

**Supervision:** Kazuhiro Wada.

**Validation:** Hongdi Wang, Kazuhiro Wada.

**Visualization:** Hongdi Wang, Kazuhiro Wada.

**Writing – original draft:** Hongdi Wang, Kazuhiro Wada.

**Writing – review & editing:** Hongdi Wang, Noriyuki Toji, Shin Hayase, Satoru Akama, Jun Sese, Kazuhiro Wada.

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
