## [Editor Report · Decision Letter 0]

23 Jul 2019

Dear Dr Wada, 

Thank you for submitting your manuscript entitled "Transcriptional regulatory divergence underpinning species-specific learned vocalization in songbirds" for consideration as a Research Article by PLOS Biology.

Your manuscript has now been evaluated by the PLOS Biology editorial staff, as well as by an academic editor with relevant expertise, and I'm writing to let you know that we would like to send your submission out for external peer review.

**Important**: Please also see below for further information regarding completing the MDAR reporting checklist. The checklist can be accessed here: https://plos.io/MDARChecklist

Please re-submit your manuscript and the checklist, within two working days, i.e. by Jul 25 2019 11:59PM.

Kind regards,

Roli Roberts

Senior Editor

PLOS Biology

INFORMATION REGARDING THE REPORTING CHECKLIST:

PLOS Biology is pleased to support the "minimum reporting standards in the life sciences" initiative (https://osf.io/preprints/metaarxiv/9sm4x/). This effort brings together a number of leading journals and reproducibility experts to develop minimum expectations for reporting information about Materials (including data and code), Design, Analysis and Reporting (MDAR) in published papers. We believe broad alignment on these standards will be to the benefit of authors, reviewers, journals and the wider research community and will help drive better practise in publishing reproducible research. 

We are therefore participating in a community pilot involving a small number of life science journals to test the MDAR checklist. The checklist is intended to help authors, reviewers and editors adopt and implement the minimum reporting framework. 

IMPORTANT: We have chosen your manuscript to participate in this trial. The relevant documents can be located here:

MDAR reporting checklist (to be filled in by you): https://plos.io/MDARChecklist

**We strongly encourage you to complete the MDAR reporting checklist and return it to us with your full submission, as described above. We would also be very grateful if you could complete this author survey:

https://forms.gle/seEgCrDtM6GLKFGQA

Additional background information:

Interpreting the MDAR Framework: https://plos.io/MDARFramework

Please note that your completed checklist and survey will be shared with the minimum reporting standards working group. However, the working group will not be provided with access to the manuscript or any other confidential information including author identities, manuscript titles or abstracts. Feedback from this process will be used to consider next steps, which might include revisions to the content of the checklist. Data and materials from this initial trial will be publicly shared in September 2019. Data will only be provided in aggregate form and will not be parsed by individual article or by journal, so as to respect the confidentiality of responses. 

Please treat the checklist and elaboration as confidential as public release is planned for September 2019.

We would be grateful for any feedback you may have.

---

## [Decision Letter · Decision Letter 1]

22 Aug 2019

Dear Dr Wada,

Thank you very much for submitting your manuscript "Transcriptional regulatory divergence underpinning species-specific learned vocalization in songbirds" for consideration as a Research Article by PLOS Biology. Your manuscript was evaluated by the PLOS Biology editors as well as by an Academic Editor with relevant expertise and by three independent reviewers.

Based on the reviews, we will probably accept this manuscript for publication, assuming that you remove the BNDF agonist injection experiments as requested by reviewer 2 and modify the manuscript to address all other concerns raised by the reviewers.

We expect to receive your revised manuscript within two weeks. Your revisions should address the specific points made by each reviewer. In addition to the remaining revisions and before we will be able to formally accept your manuscript and consider it "in press", we also need to ensure that your article conforms to our guidelines, one of which is described below under DATA POLICY and is marked with "***IMPORTANT: ". A member of our team will be in touch shortly with a set of requests. As we can't proceed until these requirements are met, your swift response will help prevent delays to publication.

Please note that you may have the opportunity to make the peer review history publicly available. The record will include editor decision letters (with reviews) and your responses to reviewer comments. If eligible, we will contact you to opt in or out.

Sincerely,

Di Jiang, PhD

Associate Editor

on behalf of 

Senior Editor

PLOS Biology

DATA POLICY:

***IMPORTANT: Regardless of the method selected, please ensure that you provide the individual numerical values that underlie the summary data displayed in the following figure panels: Figure 2BC, 3AB, 4ACD, 5ABE, 6ACD (this figure might be removed depending on how you meet the request that you must remove the BNDF agonist injection experiments), S1, S3, S4, S6, S7, as they are essential for readers to assess your analysis and to reproduce it. ***IMPORTANT: Please also ensure that figure legends in your manuscript include information on where the underlying data can be found. You may say in every relevant figure legend that, e.g., "Data associated with this figure can be found in the supplemental data file (S1 Data).".

Reviewer remarks:

Reviewer #1 (Yisi Zhang, signed review): 

In this work, Wang et al. studied the difference between two closely related but behaviorally different songbird species, the zebra finch and the owl finch, of their transcriptional regulation in the singing system. In particular, the authors investigated the cis- and trans-regulation within HVC and RA and found that the trans-regulation was more prevalent in both areas. The cis- and trans-regulations were closely related to the species-differential gene expression, especially the trans-regulated genes in RA mediated by BDNF. The latter was differentially expressed in HVC. The widespread BDNF expression levels in HVC among the F1 hybrids were significantly correlated with some song features. Over-activation of the BDNF receptor in zebra finch RA eliminated the species-specific song features and altered the expression of some genes in RA. This is an interesting and well-designed comparative study elucidating the evolutional variation of transcriptional regulation within the song production system using RNA sequencing and pharmacological manipulation. The analysis is comprehensive with sufficient statistical analysis. The arguments will be strengthened if the following issues can be clarified.

1. How different are the genomes of these two species? Since the SNPs are mapped to the ZF reference, does this method naturally introduce bias to the detection of ss-SNPs and the estimation of differential expression? Please discuss.

2. The authors hypothesized that BDNF regulates the trans-regulated genes in RA through axon terminal secretion from HVC to RA. Although the infusion of 7,8-DHF altered song structure, the change looks a lot like a song degradation as opposed to a shift to OF-like song. The hypothesis will be strengthened if there is a difference in the concentration of RA BDNF protein between the two species and/or the F1 hybrids. 

3. The authors pointed out that there is no difference in the ASE ratios of the ZO and OZ hybrids and treated them as equivalent. In Figure 6, however, there seems to be a clear division in the BDNF expression levels (at least in RA) between the ZO and OZ. I think this is worth more elaboration or discussion, especially it is interesting that there seems to be more influence from the female side (if I follow the labeling convention of figure 1). I am curious to see if there is a global difference in the song features between ZO, OZ, as well as a comparison with the ZF and OF tutor songs. 

There are some minor issues:

-Ln107, “Fig. 2”. Please specify which sub-figures one should look at. Also, it is not very clear what Fig. 2B is showing, and it is very hard for people not in the songbird field to understand the similarity matrices without explanation.

-Why these particular acoustic features were selected but not, for example, entropy, pitch or duration variance?

-Ln173, “Fig. 4B” should be “4A”.

-Ln258, “diurnal singing in HVC neurons” typo.

-Ln314, “are commonly regulated the trans-regulatory environment” typo.

-Ln402, “OF checked” typo.

-Lns901-906, figure #s are not ordered.

Reviewer #2: 

Review of “Transcriptional regulatory divergence underpinning species-specific learned vocalization in songbirds” 

This is a well-written paper form an investigator that has done important contributions to the biology of bird song. In this paper they do pioneering work as they try to identify the genetic basis that could account for differences in the types of song produced by different species.

The main strength of the paper is that the strategy is novel for songbirds. A similar approach has been used by Hopi Hoekstra for different behaviors in closely related rodent species, but for birds this is clearly the first such study. The experiments are well-performed, and the explanations are clear. My main concern is that similarly to the Hokestra papers, it is not realistic to expect to be able to pinpoint that the differences for complicated behaviors are due to changes in a single gene. Fundamentally, the final interpretation of this type of approach is that these behavioral differences are due to multiple genes interacting in complex ways. Unfortunately, some features of biology cannot be reduced to simple mechanistic models, and this work is a good example of that caveat.

My main criticism about the paper is that I do not believe that the interpretation of the BDNF receptor manipulation is correct. Fundamentally, when they inject the BDNF receptor agonist they see a “simplification” of the song of the zebra finch song – the quality of the song is degraded and it becomes less elaborate. This does not mean that it loses “species-specific acoustic and sequence traits”, it simply means that it becomes degraded. Many other experiments have song that different lesions of the song circuit produce simplified songs that are quite stereotypical. A control for this experiment would be to inject into the birds any other type of drug that affects genes that are NOT present in the genes that they identify as being differentially regulated between zebra and owl finches, and see the behavioral effects. For example, if they inject a drug that messes up a biological process not directly relevant to song production (for example, tubulin polymerization), my guess is that they will see a behavioral change very similar to what they observe with the BDNF agonist (note: they should choose a drug that is not directly toxic to the neurons, but it just messes them up in a reversible manner). 

In a more general way, I do not find the logic of this experiment very convincing. The production of song is an extremely complex process that depends on the formation of connection between different brain nuclei throughout development. The manipulation that they do in this paper involves injecting a BDNF receptor agonist in the circuit of a fully formed circuit. The specific connections between the different brain nuclei have formed during the embryonic period and during song learning. However, activating BDNF receptor in an adult animals is just “messing” with up the connections that were already there. It is interesting that the song gets degraded in a more or less predictable manner, but calling this “species-specific features is not justified. 

In summary, I think that paper is well done, and is interesting, but I do not agree that the interpretation of the BNDF experiments should be toned down drastically – BDNF receptor activation clearly changes the song, but we cannot tell that it is changing with any type of specificity. Actually, I think that the paper would be stronger without the BNDF agonist injection experiments. 

More specific comments:

. Line 209: In particular, we found that GO categories related to neural functions associated with synapse transmission, somatodendritic compartment, and nervous system development were significantly enriched for RA trans-regulated genes.  

I found this statement tautological –genes associated with“synapse transmission, somatodendritic compartment, and nervous system development” comprise a huge number of the genes expressed in neurons. Thus, it is expected that simply because of their sheer number, those genes are the ones whose pattern of expression are going to change. 

. Line 221 “ this prediction of BDNF as a trans-regulatory mediator in RA, we found two amino acid   substitutions in BDNF between ZF and OF: Ser45Arg in prodomain and Thr143Met in NGF domain (Fig. 5D)”.  

And Line 247: “In contrast, the expression level of BDNF mRNA in HVC had four  significant correlations with acoustic and sequential song parameters in F1 hybrids [acoustics: syllable duration (r = −0.862, p =0.006) and entropy vibrance (r = −0.822, p =   0.012); sequence: motif (r = −0.762, p = 0.028) and repetition (r = 0.729, p =0.040),   Pearson correlation] (Fig. 6C, D). “

I find this argument very confusing. The main type of measurements that they do are related to levels of gene expression. However, if in addition to changes in levels of expression, there are also changes in the sequence of the genes this would make it extremely difficult to separate which of these 2 factors is responsible for the behavioral changes that they observed.

Reviewer #3: 

This paper uses F1 species hybrids to investigate how transcriptional regulatory divergence between species relates to the production of species-specific songs. I generally found the question to be interesting and well-addressed. I have a few concerns, primarily about the conclusions reached with the last experiments, and well as some more minor editorial comments. In general, there are a number of places where the manuscript needs more editing for clarity and readability. 

My main concern is with the conclusion that over-activation of BDNF eliminates species-specific acoustic and sequence traits. While there are clearly changes to ZF song with over-activation of BDNF, those changes are reminiscent of simplification of song structure and increases in repeats seen with a range of other manipulations where you lose syllable complexity/diversity and increase repetition. For example, the greater stereotypy and syllable degradation you see with lesioning LMAN/DLM in young birds, or changes with deafening or with different acoustic experience during development. While a number of measures of song move in the direction of the Owl Finch song, I’m not convinced that is the same as the song becoming more like OF sing vs. just becoming simpler and noisier. It’s also not clear why changes to BDNF level would make the song more like OF song, unless it was thought that the shared ancestor of OF and ZF had a song more similar to OFs, which isn’t clear based on the songs of other grassfinch species in the clade or something discussed here. 

I am also surprised by the weak effect of learning shown in Figure 1. Because 15-25 phd is really late for removing the father, I wonder if this is why there is not a stronger effect of the cross tutoring? 

Drug infusion: 10 days is a long time for a permanently implanted microdialysis probes, were there any issues with gliosis? How was it “infused continuously during daytime”? Is it using the reservoir system? If so, Is there any data to demonstrate that there is still drug moving from the reservoir into the probe itself at the end of the day? For example, when was song recorded for the TTX controls? If it is late in the day, that could support that drug is continuing to be infused. 

Minor comments

Line 42 analog not homologue

Line 47 eliminated species-specific acoustic and sequence traits is too strong. 

Line 54-55 I’m not sure that ecology makes sense here. Not clear on what this is saying

Line 70 song system? Song pathway makes it sound especially linear

Line 80 refs are mostly about Area X

Line 91-92 this sentence is difficult to follow, rephrase

Line 92 define Gene Ontology

Line 102 Is this really a concern, that lab rearing would eliminate species-specific song features? There needs to be better justification for this.

Line 108 I’m not convinced that having somewhat overlapping values for each individual acoustic feature is an indication that there are not some physical constraints of peripheral vocal organs. Also, the sentence is confusing, I think it means to say that because the distribution ranges are overlapping, they can confirm there is not a physical constraint, but that it not how it is written on Line 109.

Line 116-120 It looks like the statistics were to compare within each tutoring condition (normal vs. cross-species). I’m not clear on what was done statistically to say that there are song tutoring effects on most of the song parameters. This would mean whether there is a difference between ZF tutored ZFs and OF tutored ZFs, correct? But none of those statistics are reported. The differences within the cross-tutored groups are convincing, and the main point of the figure, but as written this paragraph makes me wonder about the effects of tutoring. I think a statistical model with tutoring condition, species, and the interaction term would be useful here. It also seems like a lot of statistical tests (8 different song measures) to do with a very small sample size. 

Line 166-167 “we identified that over 75 and 10% of examined genes” I don’t understand what this means.

Line 183 what does “between a pair of eight F1 hybrids” mean?

Line 240-243 This statement needs to be more clear. Birds were reared hearing both ZF and OF songs? Just single examples of each? But each bird acquired an individually-unique song?

Line 245 I don’t think it’s “as a result”, delete this phrase.

Line 249 Entropy variance? Not “vibrancy”

Line 258-261 The justification for looking at BDNF needs to be explained more clearly. 

Line 314 word missing, regulated ___ in? by?

Line 342 axonal boutons, not buttons

Line 347-349 This sentence needs rephrasing

Line 361-363 Needs a better description of the AFP. The basal ganglia is homologous across vertebrates. The AFP is a specialized cortical-basal ganglia-thalamic loop

Line 369 I’m not sure that “motif structure” is a species specific trait of ZF songs. This seems to be referring to a very specific feature of motif structure, but that’s not clearly defined here.

Line 402 OF chicks not checked.

Line 581 rephrase, was infused to one hemisphere of RA.

Fig 2 Plots in 2B should be on the same scale for con and cross

Fig 6 Labels in legend are wrong relative to the figure (also out of order)

In general, there needs to be more information in the Methods about the rearing of F1 hybrids and how they were tutored.

---

## [Editor Report · Decision Letter 2]

18 Sep 2019

Dear Dr Wada,

On behalf of my colleagues and the Academic Editor, Asif A. Ghazanfar, I am pleased to inform you that we will be delighted to publish your Research Article in PLOS Biology. 

PRESS 

Kind regards,

Hannah Harwood

Publication Assistant, 

PLOS Biology

on behalf of

Roland Roberts,

Senior Editor

PLOS Biology